# Use of Nanomaterials in the Stabilization of Expansive Soils into a Road Real-Scale Application

**DOI:** 10.3390/ma13143058

**Published:** 2020-07-08

**Authors:** Julia Rosales, Francisco Agrela, José Ramón Marcobal, José Luis Diaz-López, Gloria M. Cuenca-Moyano, Álvaro Caballero, Manuel Cabrera

**Affiliations:** 1Construction Engineering Area, Rabanales Campus, Leonardo da Vinci Building, University of Córdoba, 14071 Córdoba, Spain; jrosales@uco.es (J.R.); ep2diloj@uco.es (J.L.D.-L.); manuel.cabrera@uco.es (M.C.); 2Department of Pavements, Sacyr Construction, Paseo de la Castellana, 83-85, 28046 Madrid, Spain; jrmarcobal@sacyr.com; 3Department of Building Constructions, University of Granada, 18071 Granada, Spain; gloriacumoy@hotmail.com; 4Fine and Nanochemistry, Inorganic Chemistry Department, Rabanales Campus, Marie Curie Building, University of Córdoba, 14071 Córdoba, Spain; alvaro.caballero@uco.es

**Keywords:** soil stabilization, nanomaterial, real-scale application, mechanical behavior, life cycle assessment

## Abstract

Stabilization is a traditional strategy used to improve soils with the main objective of ensuring that this base is compliant with the technical specifications required for the subsequent development of different infrastructures. This study proposes the use of commercial nanomaterials, based on a solution of silicates, to improve the technical characteristics and bearing capacity of the expansive soil. A physical–chemical property study was carried out on the additive nanomaterial. Subsequently, different mixtures of expansive soil, selected soil and artificial gravel with quicklime and commercial nanomaterials were developed to evaluate the improvement obtained by the use of nanomaterials in the technical characteristics of the soil. Compressive strength and the Californian Bearing Ratio index were considerably increased. A full-scale study was carried out in which the nanomaterial product was applied to two different sections of stabilized road compared to a control section. The results obtained showed that the use of nanomaterial led to the possibility of reducing the control section by 30 cm, thus achieving less use of quicklime and a mechanical means for preparing the road section. The use of commercial nanomaterial improved the behavior of the stabilized sub-base layer. Through life cycle assessment, this study has shown that the use of nanomaterials reduces the environmental impact associated with soil stabilization.

## 1. Introduction

The use of stabilization techniques has increased significantly in recent decades due to new construction sites, which are increasingly located in poor quality land areas. Soil improvement will be critical in future geotechnical practices to adopt cost-effective solutions to achieve reductions in the amount of material used. In regions where groundwater intrusion problems exist, altered permeability is often a major factor in soil stabilization.

In civil engineering, soil stabilization is mainly used to improve mechanical properties, bearing capacity and stability, as well as to improve the permeability, plasticity and durability of the soil. In addition, the use of stabilizing products chemically modifies the soil to improve its characteristics [1].

It has been demonstrated that the chemical stabilization of soils, in particular of expansive clays, leads to improving their properties to be considered as a structural layer of roads [2,3]. Previous studies have shown the use of lime and Portland cement to be the most widely used technique among the wide range of materials used for soil stabilization [4,5,6,7]. However, the use of these traditional materials has certain disadvantages, such as high CO_2_ emissions in their production, high energy consumption and high cost. In addition, they can damage certain soil properties and, with their use, it is difficult to reduce the section of soil stabilized to comply with the required technical properties.

As an alternative to traditional stabilizing materials, it is increasingly common to study the use of by-products, such as fly ash [8,9], biomass bottom ash [10,11] and phosphogypsum [12,13,14]. The use of different types of slag—mainly steel slag—has also been studied [15,16,17].

As an alternative for soil stabilization, in recent years, nanomaterials have attracted considerable scientific interest because they can actively interact with expanding clay soil particles, mainly because of their high specific surface area [18]. The types of nanoparticles most commonly used in cementitious compounds are SiO_2_, TiO_2_, Al_2_O_3_ and carbon nanotubes [19].

Among the most common chemical agents for stabilization, apart from lime and cement, numerous previous investigations have studied the application of sodium silicate, considering this method of stabilization as non-traditional. Soil stabilized with sodium silicate improves the resistance of the soil [20]. However, when it is applied in powder form, it is difficult to insert into the pores of the soil and more difficult to apply in situ [21]. For this reason, the use of nanomaterials is more effective—of all nanomaterials, nano-SiO_2_ has a high pozzolanic capacity due to its pure amorphous SiO_2_ composition that characterizes it as a highly potential material for soil stabilization [1,22]. The use of nano-SiO_2_ particles in combination with a reduced percentage of lime or cement leads to the modification of the expansive soil properties, because these particles are the result of a chemical reaction between SiO_2_ and Ca(OH)_2_ during the hydration of the cement or lime [22,23]. The use of nano-SiO_2_ for soil stabilization has a significant influence on the microstructure and the physical and chemical properties of soils [24], in addition to improving their compaction density [25,26,27]. In combination with cement it considerably improves the geotechnical properties of soils [28]. According to different studies [28,29] the use of nano-SiO_2_ shows an increase in the unconfined compressive strength of the soil in relation to the use of other traditional stabilizers, such as lime or cement.

The study of nano-silica has not only been carried out in soil stabilization. Previous studies show its possibility for use in asphalt concrete pavements, demonstrating an improvement in ageing properties, resistance and rheological properties [30,31].

In this study, a nanomaterial of Sodium Silicate-Based Admixture (SSBA) was applied for soil stabilization in combination with lime and a vinyl acetate homopolymer coating (VAH).

In this research, the physical properties and mechanical behavior of soils stabilized with SSBA were studied for real application in the structural layers of rural roads. The full-scale application of the nanomaterial was performed by building a section of the road, using SSBA in two sections in combination with expanded clay, artificial gravel and selected soil.

Through the life cycle assessment (LCA) performed, the possible environmental impact reduction associated with the use of nanomaterials as a viable and sustainable solution in soil stabilization was verified.

## 2. Experimental Program

The work comprises four differentiated parts (Figure 1): A basic characterization of the material used to establish its properties (expansive clay soil, selected soil and nanomaterial); A second part in which different mixes of soil and nanomaterial are evaluated in the laboratory, by means of these tests the bearing capacity of the soil is checked. The tests carried out are those required by the Spanish PG-3 regulations for the evaluation of soils for the subsequent construction of traffic routes; A third test, in which a real road section is built using nanomaterial, facilitates the reduction in the thickness of the section. The properties acquired by the section are studied by means of deflection tests and a loading and unloading plate, which are the two most common methods for evaluating a section by means of non-destructive tests; Finally, to evaluate the environmental impact caused by the execution of the real section alternative to a traditional construction solution with nanomaterials, a life cycle analysis is developed, which shows the reduction in consumption and CO_2_ emissions.

The studies carried out in each of the stages are described in detail below.Stage 1.Study of materials—In this phase the physical–chemical properties of the soil used, the quicklime and the nanomaterial were evaluated.Stage 2.Laboratory study of soil mixes with nanomaterial—Laboratory study comparing the stabilization of an expansive soil with only lime and the addition of SSBA with the application of a synthetic cover product (vinyl acetate homopolymer in aqueous dispersion) to improve the bearing capacity and mechanical properties of the soil. The application rate of SSBA was 0.65 L/m^3^ of soil. The bearing capacity and mechanical behavior parameters of each of the mixtures were evaluated in the laboratory.Stage 3.Dimensioning of the pavement package—Execution of experimental section and monitoring of the road section in the short and medium term. Based on the results obtained, the design and dimensioning of three pavement packages was carried out. A first control section was designed with soil, stabilized with only lime, a second alternative section with the application of SSBA and a combination of expansive soil with artificial gravel and a third section combining expansive soil, selected soil and SSBA. The three mixtures were made in order to find the optimum section by reducing thickness and improving the bearing capacity and mechanical behavior of the pavement. Subsequently, an experimental section was carried out and the execution of the work controlled by verifying that the previously established specifications were met and corroborated by laboratory tests. The road section was executed in Villacarrillo (Jaen), Andalusia, Spain. Once the execution of the projected sections was completed, short and medium term monitoring tests were carried out to evaluate the basic parameters.Stage 4.Life cycle analysis—In this phase, the environmental impact caused by the construction processes of the three sections of stabilized soil projected is analyzed.

## 3. Materials

### 3.1. Soils

In this study three types of soils were analyzed for their subsequent stabilization and improvement through the application of lime and SSBA. These soils were expansive clay soil (ECS), artificial gravel (AG) and soil from the rejection of gravel manufacture, also called selected soil (SS).

Expansive clay soil (ECS): Expansive soils are those that show volumetric changes in response to changes in their moisture content. These soils swell when the moisture content increases and contract when the moisture content decreases. The expansive clay soil that is analyzed comes from the plot of land where the real-scale study was conducted, in Villacarrillo, Jaen, Spain. These clays are mainly characterized by high calcite content and a high plasticity index.

Artificial gravel (AG): This is a mixture of aggregates, totally or partially crushed, in which the particle size distribution of all the elements that compose it is of a continuous type. Artificial gravel was produced in a mobile production plant, located in Villacarrillo (Jaen), Andalusia.

Selected soil (SS): The soil selected is a mixture of arid limestone from the crushing of natural quarry stone and comes from the same quarry as the artificial gravel. This soil is classified according to the General Technical Specification for Construction of Roads and Bridges PG-3 [32] as Selected Soil.

The physical and chemical properties of ECS, AG and SS are shown in Table 1.

From the liquid limit test, the non-plasticity of the AG is verified; therefore, its plasticity index is non-plastic. However, ECS presents a plasticity index close to 30%.

In the case of SS, the prescription of this physical characteristic of the material, in accordance with that of PG-3 (General Technical Specification for Construction of Roads and Bridges) [32], classifies as selected soil that have a plasticity index of less than 10%. In addition, the content of organic matter is less than 0.2%, so it complies with current technical specifications.

As shown in Table 1, ECS, AG, and SS are primarily composed of Ca and Si.

Figure 2 shows the particle size distribution of the soils. ECS presents a particle size distribution with a majority presence of fine particles, with a percentage of passage through the sieve of 0.063 mm of 96.7%. The maximum particle size that is retained is 5 mm. AG presents a continuous particle size distribution with a maximum aggregate size of 20 mm and a percentage of fines of less than 9%. AS is mainly made up of thick aggregates with a maximum size of 32 mm, with a fine particles percentage of less than 6%.

A mineralogical study of the three types of soils used in this work was carried out. The methodology used was powder analysis by X-ray diffraction with Bruker equipment (model D8 Advance A25) and a Cu twist tube. DifracPlus EVA 3.1 software was used to identify each of the crystalline phases of which the soils are composed.

Figure 3 shows the mineralogical composition of each soil type.

It was observed that ECS is mostly composed of calcite, silicon, quartz and dolomite. However the AG and SS showed a main composition of dolomite.

The presence of carbonates (calcite and dolomite) is abundant in soils of the Guadalquivir River, southern Spain, according to previous studies [33]. Quartz was the second-most abundant mineral.

### 3.2. Quicklime

Hydrated lime is obtained when quicklime (CL 90-Q) reacts chemically with water. Hydrated lime (calcium hydroxide) reacts with clay particles and permanently transforms them into a strong cementitious matrix.

### 3.3. Sodium Silicate Based Admixture (SSBA) and Vinyl Acetate Homopolymer (VAH)

A liquid solution was considered, based on nanosilica in a concentrated version (stock solution) called SSBA, used to stabilize the soil together with lime. Additionally, VAH was applied in aqueous dispersion to cover the stabilized soil surface. Its purpose is to improve the waterproofing of the surface.

First a Thermogravimetric Analysis (TGA) of the liquid solution of the nanomaterial was carried out. Subsequently, the chemical properties X-Ray Fluorescence (XRF), made with Primus IV, Maxxi 6 and Micro Z equipment, and the XRD of the solid powder fraction remaining after solvent removal (i.e., the resulting nanomaterial fraction after TGA), were analyzed.

These studies were carried out for a better understanding of the material and how its application with different types of soils can improve load capacity and mechanical properties. 

The results of TGA analysis are shown in Figure 4. The progressive weight loss to 200 °C is due to the loss of water in the nano-silane. SSBA showed high silica content, containing 32%. The thermal behavior of silicates in the presence of water has been reported and is in agreement with the results obtained. According to the TGA, once it achieves a temperature of 200 °C, the weight of the material remains stable until the end of the test (at 800 °C). Only a minimal weight loss (1–2%) is barely detectable in this range. This fact confirms that the material that the solute making up the solution is a silicon-based compound that does not oxidize or decompose in this temperature range. The thermal behavior of silicates in the presence of water has been reported and is in agreement with our observations [34].

The main weight loss of the sample starts at 48 °C and progresses to 148 °C, at which temperature the weight loss stabilizes to the stable SSBA mass content of 32%.

The high Si percentage of the nanomaterial, as shown in Table 2, in combination with the Ca present in soils and lime can produce calcium silicate or aluminum silicate hydrates in the polymerization process [35]. The combination of both elements to generate these hydrates is carried out through hydration mechanisms similar to those produced in Portland cement [36]. For this reason, the hardening and greater stabilization of the soil is achieved.

Sodium Silicate-Based Admixture DRX showed that the nanomaterial is mainly composed of dolomite, quartz and sodium silicate, as shown in Figure 5. Previous studies have demonstrated the advantages of sodium silicate as a soil stabilizer. The stabilization of montmorillonite-rich clay soils with sodium silicate powder and lime achieved high mechanical properties [20,37,38].

## 4. Laboratory Tests, Methods and Results

In this section, the results of the laboratory tests that were carried out on the mixtures that formed the different layers of the road sections are shown. The Modified Proctor compaction test, Californian Bearing Ratio index (CBR) and unconfined compressive strength were determined.

### 4.1. Mix Design

The soils analyzed were combined with each other in different percentages—quicklime and SSBA were also added to obtain the mixtures used in the sections of the road. Three specimens of each of the mixes were manufactured to carry out the tests. Table 3 shows the mixtures that were analyzed in the laboratory and the nomenclature used for them.

The mixtures were manufactured in the laboratory, emulating the real conditions of road section construction and, therefore, quicklime was added to the total dry mass to stabilize it. Likewise, SSBAs were added to the total dry volume of soil to stabilize it. The dosages are expressed in weight per cubic meter of material. The mixes AG/ECS (50/50) + 1.5% QL + SSBA and SS/ECS (50/50) + 1.5% QL + SSBA resulted from mix 0.50 m^3^ of ECS with 0.50 m^3^ AG and 0.50 m^3^ of ECS with 0.50 m^3^ SS, respectively, plus quicklime and SSBA.

### 4.2. Proctor Test/Moisture-Density Relationship

The Modified Proctor (MP) was made according to the UNE EN 103501: 1994 standard. This test—similar to the standard Proctor but, in this case, with five layers of soil compacted, applying higher compaction energy—is used to determine the compaction properties of the soil; specifically, the optimal moisture content added to the soil in order to reach the maximum dry density.

Moisture-density curves provide information about the compaction sensitivity of materials with changes in moisture content [39]. Materials with flat curves show less variations in dry density with moisture changes. However, materials with shaped curves present higher variations in dry density caused by small changes in moisture, making it necessary to ensure a water addition close to optimal moisture content to obtain the maximum dry density.

The moisture content–dry density curves of AG, ECS and the four mixtures are shown in Figure 6.

The moisture–density curves are related to the physical properties of the materials. According to the results shown in Table 4, AG presents the highest maximum dry density and lower optimal moisture content, due to the larger particle size and higher SSD density than ECS, as shown in Figure 2 and Table 1. In contrast, ECS, ECS + 1.5% QL and ECS + 1.5% QL + SSBA present the lowest maximum dry density values and the highest moisture contents, but it can be observed that the addition of SSBA results in an increase in a dry density and a decrease in the optimal moisture content. Unlike the use of traditional stabilizers, such as lime or cement, which leads to a decrease in density and an increase in the optimum moisture content [40]. Finally, the SS/ECS (50/50) + 1.5% QL + SSBA and AG/ECS (50/50) + 1.5% QL + SSBA materials present intermediate values due to the combination of soils, observing that the combination with SS presents a flat curve due to the higher percentage of total fines, in contrast to the combination with AG that presents a shaped curve.

### 4.3. Vibrating Hammer Times

The compaction of the materials for CBR and compression strength tests was carried out using a vibratory hammer in accordance with NLT-310/90 standard.

Each mold is filled with three layers and each layer is formed with a thickness of approximately one-third of the length of the mold and it is compacted applying the same time to each layer. The operation is repeated three times, modifying the compaction times. After all three repetitions, the hammer time density diagram is obtained.

From the density–compaction time curve, the time required for the compaction of the sample of at least 98%, according to the Modified Proctor, is obtained, as observed in Figure 7. Compaction times range from 23 to 27 s.

### 4.4. California Bearing Ratio (CBR) and Compressive Strength

The CBR test method measures the bearing capacity of a soil or compacted aggregates in a laboratory under controlled density and moisture conditions and is used to assess the suitability of soil as a sub-grade, sub-base or base for structural road layers, as well as for the classification of soil, as indicated by authors such as Nagaraj et al. [41].

The CBR value is carried out in accordance with UNE 103-502, which describes the process for determining the resistance index of soils, called CBR. This index is not an intrinsic value of the material but depends on the density and moisture conditions of the soil and the overload that is applied when the test is carried out.

The compressive strength test is carried out to determine the shear strength of cohesive soil or mixtures treated with hydraulic binders. This test was done according to the NLT-305/90 standard.

The value of simple compression strength depends on several factors, such as the type and amount of hydraulic binder, the efficiency of compaction, the type of soil, etc. In the execution of road layers, the values of simple compression strength in mixtures treated with cement determine the types of layer and road that can be built. The soils stabilized with quicklime have no limitations in the Spanish regulations in values of simple compression strength; even so, it is a value that shows the mechanical behavior of the road layer in a complementary way to the CBR index.

CBR and compressive strength tests were carried out on compacted samples with the optimal moisture content, according to the Modified Proctor test. The CBR test was carried out in seven-day soaked conditions and the compressive strength test was performed with a seven-day cure in a wet chamber. The results are shown in Figure 8.

AG is a material that presents a wide range of bearing capacities that may present values between 30–180% CBR. On the contrary, expansive soils, such as clays, are considered non-usable soils, according to Spanish regulations.

Analyzing the results shown in Figure 8, it can be observed that AG is medium quality gravel, and ECS had insufficient geotechnical quality.

The addition of lime and lime plus SSBA greatly improves the bearing capacity of the soil, allowing for its use in road layers.

The mixtures SS/ECS (50/50) + 1.5% QL + SSBA and AG/ECS (50/50) + 1.5% QL + SSBA show an adequate mechanical behavior, with typical CBR values of AG, in addition to mixtures showing values of 0.51 and 0.60 MPa of simple compressive strength, improving mechanical behavior.

The use of other types of polymers in previous studies also led to an improvement in the soil’s CBR index [42], however in only 30% increases in the CBR index were achieved with respect to unstabilized soil. In this work the soil bearing capacity was improved by 170%.

The use of nano-silane in conjunction with lime improved the compressive strength of the expansive clay. Similar results were obtained in previous studies [43,44] in which it was observed that nanosilica alone had no improvement effect on short-term mechanical behavior. However, the pozzolanic reaction, produced by the silica nano-particles combined with the lime, increased the strength significantly.

## 5. Road Layer Design: Methods and Results

An experimental section composed of three different sections, each one approximately 160 meters long, was carried out (Figure 9).

The study was based on the design of the pavement package for the Villacarrillo-Villanueva del Arzobispo section (Jaén, Spain) of the A-32 Linares-Albacete highway.

### 5.1. Execution Process and Sections

Three different sections were developed:-Control Section: Control section composed of three layers. Two 25 cm thick layers of expansive clay soil, stabilized with 1.5% quicklime (ECS + 1.5%QL) and a 30 cm thick layer of artificial gravel (AG).-Alternative section 1: Section applying the use of nanomaterials to improve soil stabilization properties. A reduction of 30 cm in thickness is achieved with respect to the control section. This section is formed by two layers: A 25 cm layer of expansive clay soil, stabilized with 1.5% quicklime and with SSBA (ECS + 1.5%QL + SSBA). SSBA is applied within the water mixture necessary to reach optimum humidity in a 1:32 solution with a 0.65 L/m^3^ ratio; a 25 cm finishing layer of stabilized expansive clay soil combined with artificial gravel in proportions of 50%/50% with 1.5% quicklime and with SSBA (ECS/AG + 1.5%QL + SSBA). VAH was applied to the finished section as a covering material.-Alternative section 2: Section similar to Alternative section 1, where the use of nanomaterials is applied, and a reduction of 30 cm thickness is achieved with respect to the control section. Two layers make up this section—a 25 cm layer of expansive clay soil, stabilized with 1.5% quicklime and with SSBA (ECS + 1.5%QL + SSBA) and a 25 cm finishing layer of stabilized expansive clay soil combined with artificial soil in proportions of 50%/50% with 1.5% quicklime and with SSBA (ECS/SS + 1.5%QL + SSBA). VAH was applied to the finished section as a covering material.

Figure 10 shows an illustration of each of the sections developed.

### 5.2. Execution Evaluation

For the correct application of the material, it is necessary to determine the density and moisture content of the soil in situ and check that it complies with the densities determined in the laboratory to reach the values established in the modified Proctor test. This study was carried out in accordance with ASTM D-6938.

Density and moisture content measurements were taken for each of the executed layers. This test method is a rapid non-destructive technique used as an acceptance test for compacted soil layers.

The values obtained are shown in Table 5.

The density values obtained on site were between 97% and 99% of the densities obtained in the laboratory by means of the Proctor test.

The use of SSBA resulted in a reduction in the optimum moisture values.

### 5.3. Plate Loading Test

The plate bearing test (or plate loading test) is an in situ load bearing test of soil used for determining the ultimate bearing capacity of the ground and the likely settlement under a given load.

The plate bearing test is carried out in accordance with NLT 357/98. It basically consists of loading a steel plate of known diameter and recording the settlements corresponding to each load increment. The test load is gradually increased until the plate starts to settle at a rapid rate. The total value of load on the plate, divided by the area of the steel plate, gives the value of the ultimate bearing capacity of soil.

Plate loading tests were carried out on two different dates to study the evolution over time. The first test was carried out in November 2018 and the second test was carried out in October 2019.

The results of the plate loading tests show us the Young module (Ev). A test device with a 200 kN plate support was used, and Young’s modulus was calculated at the first plate load (Ev_1_) and a second plate preload (Ev_2_), using a 300 mm diameter steel plate. Figure 11 shows the evolution of the load capacity of the three test sections.

The figure shows Young modulus Ev_2_ and Ev_1_ and, as is visible, the sub-grade had good capacity in all sections, and they showed similar behavior in all sections. It can be observed that, in addition to maintaining a good load capacity, this increases with time. In the particular case of the control section, it had a notable increase in October (2019) compared to November (2018), due to the circulation of heavy vehicles on that section.

Table 6 shows the values of the relationship Ev_2_/Ev_1_ for each section. All the values are within the limit established by the Spanish regulations [32], which establishes a maximum value of 2.2 for the different traffic categories. The control section presents values that exceed the established limit, possibly due to high compaction caused by vehicular traffic.

### 5.4. Deflection Measurements

The impact deflectometer (FWD) is a piece of equipment used to assess the structural condition of the pavement. A Dynatest Heavy Weight Deflectometer 8081, equipped with seven geophones, was used. The geophones were located at 0–300–450–600–900–1200–1500 mm. The test consists of applying a load on a test plate 30 or 45 cm in diameter and measuring the deformation produced on its surface by its effect. The variation range of said load is between 7 and 124 kN. Deflections are measured using seven sensors (geophones) that are located one below the load plate and the other six at variable distances of up to 2.9 m from the point of impact.

Using the FWD, surface data were taken every 10 m on two different dates in January (2019) and September (2019), respectively. Figure 12 shows the data obtained in each of the sections studied.

It can be observed that all the sections improved over time, decreasing their deflections. The mean deflection is shown with a black horizontal line for the test carried out in January, and a red horizontal line for the test carried out in September. In the case of alternative section 1, its deflection improved by approximately 40%, very similar to the control section, although the deflection obtained was less than the rest of the sections (118 MPa in January and 68 in September), mainly due to the continuous passage of heavy vehicles.

A methodology has been used to evaluate the structural capabilities of the pavement from the deflections measured on the surface of the different layers, called the inverse calculation method.

Inverse calculation or analysis is considered a fundamental application of FWD equipment. It consists of determining a road structure, defined by the layer thicknesses and the structural characteristics of the materials that compose it, which, when subjected to a load equal to that generated by the impact deflectometer, resemble real theoretical data. The procedure used to predict them is based on the applied dynamic load, the radius of the load plate, the thickness of the layers and the Poisson coefficients of the materials that make them up. It is an iterative method and is solved with the help of the software. In this study, the software used is EVERCALC (Washington State Department of Transportation, 2005)

According to the results shown in Table 7, a decrease in the obtained values of the elastic modulus in the sub-base can be seen. All the sections present the same patterns, relative to the elastic modulus, except the control section, whose modulus values are higher mainly due to over-compaction because of vehicular traffic.

## 6. Life Cycle Assessment

In this section, the life cycle assessment (LCA) was applied to the execution process of the three different sections developed: Control Section; Alternative section 1; Alternative section 2.

Life Cycle Assessment (LCA) allows for the quantifying of the environmental impacts of a product, process, or system throughout its life cycle. According to ISO 14040 [45] and ISO 14044 [46], the LCA application is based on four stages: definition of the goal and scope; inventory analysis; life cycle impact assessment; interpretation of the results. In this LCA study the first two stages are included in this section, while the remaining two are collected in the “Life cycle impact assessment” and “Discussion” sections.

### 6.1. Life Cycle Assessment Goal and Scope Definition

LCA study was carried out to determine and compare the environmental impact associated with the execution of the three experimental sections developed in Section 5.1 (Control Section, Alternative section 1 and Alternative section 2).

The functional unit (FU) was a road section 120 m long and 8 m wide with a thickness of 80 cm for the Control Section, and a thickness of 50 cm for Alternative section 1 and Alternative section 2.

Figure 13 shows system boundaries for Control Section, shown in Figure 13a, for Alternative section 1, shown in Figure 13b, and for Alternative section 2, shown in Figure 13c. The system boundaries include the material production (artificial gravel, selected soil, expansive clay soil, quicklime, nanomaterials and water) and the transport and execution of the road section, that is, the LCA is limited from cradle to gate. In details, the system boundaries include the following steps:The production of the raw materials: artificial gravel, selected soil, expanded clay soil, quicklime, water, SSBA and VAH.Transportation of materials from their place of production to the construction site of sections. The following transport distances were considered: (i) 5.5 km for artificial gravel and selected soil; (ii) 0.1 km for expanded clay soil; (iii) 308 km for quicklime; (iv) 0.1 km for water; (v) 341 km for nanomaterials SSBA and VAH.The construction of the three experimental sections through the following activities:

Control Section: For layer 1, the rotavator was used to scarify and prepare the ground, giving rise to expansive clay soil. Subsequently, a panning tractor extended quicklime and it was mixed with expansive clay soil using a rotavator. Then, irrigation was carried out by means of a tank truck and the levelling of the ground with a motor grader. Finally, the compaction process was completed by means of a compactor. For layer 2, expansive clay soil was transported from a collection area on site and distributed over the section using the scraper. The quicklime was spread with the panning tractor and it was mixed with expansive clay soil using the rotavator. After that, irrigation, levelling and compaction were carried out as with layer 1. For layer 3, artificial gravel was transported in trucks from the quarry to the place of execution of the section and extended by the motor grader. Subsequently, the irrigation, levelling and compaction of the layer were carried out.

Alternative section 1: For layer 1, the rotavator generated expansive clay soil and mixed it with the extended quicklime, as in Control Section. Subsequently, irrigation with water and SSBA, levelling and compaction were carried out. For layer 2, the expansive clay soil was transported from the collection area and spread over the section, the quicklime was extended and both materials were mixed with the rotavator. Then, irrigation with water and SSBA, levelling and compaction were carried out. For layer 3, an irrigation of VAH was applied by tank truck.

Alternative section 2: Layer 1 was carried out in the same way as for Alternative section 1. For layer 2, selected soil and expansive clay soil were deposited and spread on the section. Then, the quicklime was spread, and all the materials were mixed with the rotavator. Subsequently, irrigation with water and SSBA, levelling and compaction were carried out. For layer 3, the irrigation of VAH was applied by a tank truck.

The impact assessment was conducted for the categories: abiotic depletion of elements (ADe); abiotic depletion of fossil fuels (ADf); global warming (GW); ozone layer depletion (ODP); human toxicity (HT); fresh water aquatic ecotoxicity (FWAE); marine aquatic ecotoxicity (MAE); terrestrial ecotoxicity (TE); photochemical oxidation (POF); acidification of soil and water (A); eutrophication (E). For these categories, the characterization factors of the CML-IA (v 4.7) method [47] were used, as established in EN 15804 + A1 [48], regarding the sustainability of construction sites for construction products and services.

The data collected during the inventory phase were loaded into the SimaPro 9.0.0.49 software [49] and processed using the CML-IA (v 4.7) method [47]. Initially, the impact values associated with the construction of the designed sections were determined. Subsequently, the contribution of the processes was defined in order to identify those that generate the greatest impact.

### 6.2. Life Cycle Inventory

The development of the life cycle inventory was carried out by compiling the main inputs and outputs of all the processes included in the boundaries of the system. Primary data were site-specific and collected through interviews with producers and experts involved in construction sections. As secondary data for generic materials, energy and transport, the Ecoinvent v3.5 database (cut-off) [50] was used.

The following aspects were considered for the creation of the inventory:Inventories on the artificial gravel and selected soil were determined, according to the production processes, by means of primary data and Ecoinvent v.3.5 database (cut-off) [50]. Both materials were produced at the Villacarrillo quarry, located 5.5 km from the place of the construction section. The material was extracted by blasting and, after several crushing and screening processes, artificial gravel was obtained. The selected soil came from the rejected material of gravel manufacturing.The expansive clay soil comes from the plot of the land where the real-scale study was conducted. The production process was carried out by means of a loader, which extracted the material and deposited it in the collection area at a distance of 0.1 km.Inventories for quicklime and water were created from the processes of the Ecoinvent v.3.5 database (cut-off) [50].Inventories for nanomaterials (SSBA and VAH) were developed from the data provided by the manufacturer and according to the process described by Roes et al., 2010 [51].

The amount of raw materials, the machinery and its operating time and transport distances of the raw materials are shown for each section in the functional unit in Table 8.

### 6.3. Life Cycle Impact Assessment

The characterization results for the construction of the three sections are shown in Table 9; likewise, the percentages of variation of the characterization values of the alternative sections with respect to the value of the control section for each of the impact categories, are indicated. The Control Section reached the highest values in all categories except for the abiotic depletion of elements (ADe), photochemical oxidation (POF) and global warming (GW). For these categories, Alternative section 1 generated the highest impact values, with an increase of up to 24.35% for ADe, over the Control Section. In contrast, Alternative section 2 showed the lowest impact values in all categories except for ADe and POF; reductions ranged from 2.86% for GW to 33.09% for marine aquatic ecotoxicity (MAE), with respect to the Control Section.

The fact that Alternative section 1 and Alternative section 2 have a greater impact than the Control Section in categories ADe and POF (and in Category GW in Alternative section 1), may be due to the greater amount of quicklime present in the sections to stabilize the soil. Likewise, the reduction in impact observed in the other categories is due to the lower amount of artificial gravel, expansive clay soil and water required for the execution of the alternative sections.

The contribution to the impact for each category is represented for each section in Figure 14. For Control Section, shown in Figure 14a, quicklime generated the highest impacts in eight of the eleven categories analyzed (ADf, GW, ODP, HT, TE, POF, A, E), with contribution percentages ranging from 16.66% for ADe to 75.9% for GW. These results were produced as a consequence of the large amount of air emissions of carbon dioxide fossil generated during its manufacture, due to the process of the calcination of limestone at high temperatures and the energy required. Likewise, the production of artificial gravel generated the highest impacts in the MAE (47.81%) and FWAE (44.86%) categories, as a consequence of the air emissions of hydrogen fluoride, and the water emissions of Beryllium and Nickel, derived from the blasting process. For the ADe category, the transport of quicklime caused the greatest contribution (36.5%), due to the consumption of cadmium and lead related to the transport processes. The remaining processes showed the following percentages of contribution to their impact: (i) for the transport of artificial gravel, they oscillated between 5.53% (GW) and 23.39% (ADe); (ii) for the production of expansive clay soil, less than 3.33%; (iii) for the transport of expansive clay soil, below 4.05%; (iv) for the production of water, less than 6.8%; (v) for the transport of water, below 0.59%; (vi) for the execution of Control Section, less than 1.93%.

For Alternative section 1, shown in Figure 14b, the process that contributed most to the impact in all categories except ADe was quicklime, with percentages ranging from 14.58% for ADe to 81.83% for GW. The transport of quicklime caused the largest contribution in the ADe category (31.94%), and was also responsible for the second largest contribution in the categories (ADf, GW, ODP, POF, A and E), with percentages ranging from 4.93% in POF to 15.83% in E. The production of artificial gravel caused the second largest contribution in the FWAE, MAE and TE categories, with impacts ranging from 3.59% for POF to 23.27% for MAE, while its transport caused between 1.97% for POF and 7.82% for E. Regarding the contribution to the impact of nanomaterials, the manufacture of SSBA generated the second highest contribution in the ADe category (20.17%), while for the remaining categories its contribution was lower than 7.3%. Regarding the manufacture of VAH, the impact generated ranged from 1.9% for GW to 11.35% for ADe. The transport processes associated with SSBA and VAH caused impacts of less than 3.8% and 1.7%, respectively. For the remaining processes, the percentages of contribution to impact were: (i) in the production of expansive clay soil, less than 1.98%; (ii) in the transport of expansive clay soil, less than 2.15%; (iii) in the production of water, less than 5.23%; (iv) in the transport of water, less than 0.31%; (v) in the execution of Alternative section 1, less than 1.75%.

Regarding the contribution to the impact of the processes that make up Alternative section 2, shown in Figure 14c, and similar to Alternative section 1, quicklime generated the highest impacts in all categories except for ADe, and the contribution percentages ranged from 14.75% for ADe to 83.89% for GW. The transport of quicklime caused the largest contribution in the ADe category (32.32%), and was also responsible for the second largest contribution in the categories (ADf, GW, ODP, HT, FWAE, TE, A and E), with percentages ranging from 5.96% in GW and 17.73% in E. Regarding the contribution to the impact of nanomaterials, SSBA generated the second highest contribution in the ADe category (20.69%), while for the rest of the categories its contribution was lower than 8.97%. Regarding VAH, the impact generated caused the second highest contribution in the MAE (12.83%) and POF (7.67%) categories, and for the remaining categories it oscillated between 13.24% for FWAE and 1.97% for GW. The transport processes associated with nanomaterials caused impacts of less than 3.9% and 1.71%, respectively. The production of selected soil generated impacts of less than 5.18%, and its transportation process less than 8.04%. For the remaining processes, the percentages of contribution to the impact were: (i) in the production of expansive clay soil, less than 2.25%; (ii) in the transport of expansive clay soil, less than 2.22%; (iii) in the production of water, less than 6.21%; (iv) in the transport of water, less than 1%; (v) in the execution of Alternative section 2, less than 1.8%.

### 6.4. Life Cycle Assessment Discussion

According to the results obtained, the environmental impacts generated by Alternative section 1 and Alternative section 2 were lower in almost all categories as a consequence of the lower amount of artificial gravel, expansive clay soil and water required in its composition. For the Control Section, the impacts generated in the ADe and POF categories were the lowest, due to the lower amount of lime used in the soil stabilization process. In fact, the contribution analysis determined that quicklime was the process that contributed the most to the generation of impact, regardless of the section evaluated. Similarly, artificial gravel also made a significant contribution in Control Section but less so in Alternative section 1, when used in combination with expansive clay soil. Regarding the incorporation of nanomaterials, the contribution to the impact of SSBA and VAH was less than 20% and 13%, respectively. For the rest of the components of the sections (i.e., expansive clay soil, selected soil and water) the contribution was less than 7%, while for the processes of transport of the raw materials, it was less than 8%.

Table 10 shows the characterization results for the GW category (kg CO_2_ eq.) for each section according to the components of the layers.

The values obtained for the layer formed by expansive clay soil and quicklime are similar in three sections—around 6180 kg CO_2_ eq. However, the partial replacement of expansive clay soil by artificial gravel or selected soil increases the CO_2_ eq., being higher when using artificial gravel (Alternative section 1) with 8357.8 kg CO_2_ eq., compared to 6601.8 kg CO_2_ eq. (Control Section). As for the incorporation of nanomaterials in the formation of the Alternative Sections, SSBA generates 279 kg CO_2_ eq., and VAH produces 323 kg CO_2_ eq.—these values constitute about 4% of the total impact of the Alternative Sections.

## 7. Conclusions

In this study an analysis was developed on the influence of the application of a nanomaterial on the stabilization of expansive clay soils. A laboratory study was carried out and, in a novel way, a section of rural road was built to real scale, evaluating its characteristics by short-term and medium-term tests.

The following conclusions have been obtained from the results:-The nanomaterial is mainly composed of silica in the form of quartz and sodium silicate. The combination with the calcium present in the soil and the lime produces a hardening of the mixture and leads to an increase in the resistance of the soil.-The addition of SSBA to expansive clay soil with lime increases the CBR index by 50%. The use of nanomaterial leads to an improvement in the mechanical behaviour of the soil mixtures analysed, resulting in an increase in the CBR index and the compressive strength. Through the use of nanomaterials, the mechanical properties of the pavement are improved so that it is possible to execute two sections with a 30 cm reduction in thickness in comparison to the control section.-The three sections of segments executed improved over time, showing a reduction in deflection values.

All sections show the same patterns, relative to the elastic modulus, except for the control section, whose modulus values are higher mainly due to over compaction by road traffic.

Although section reduction by the use of nanomaterials led to a reduction in the elastic modulus with respect to the control, the values of the E_v2_/E_v1_ ratio of each section are within the limit established by Spanish standards.
-The life cycle assessment developed in this study has shown that the use of nanomaterials to improve soil stabilization properties reduces the environmental impact associated with the construction of the section in almost all the categories evaluated, so it is presented as a viable alternative.

Therefore, it can be concluded that the use of nanomaterials leads to an improvement in the stabilized soil properties, increasing the CBR index value and the compressive strength. This improvement in the properties makes a reduction in the thickness of the layers possible, achieving similar behaviors as in the project solution but with a lower quantity of material.

## Figures and Tables

**Figure 1 materials-13-03058-f001:**
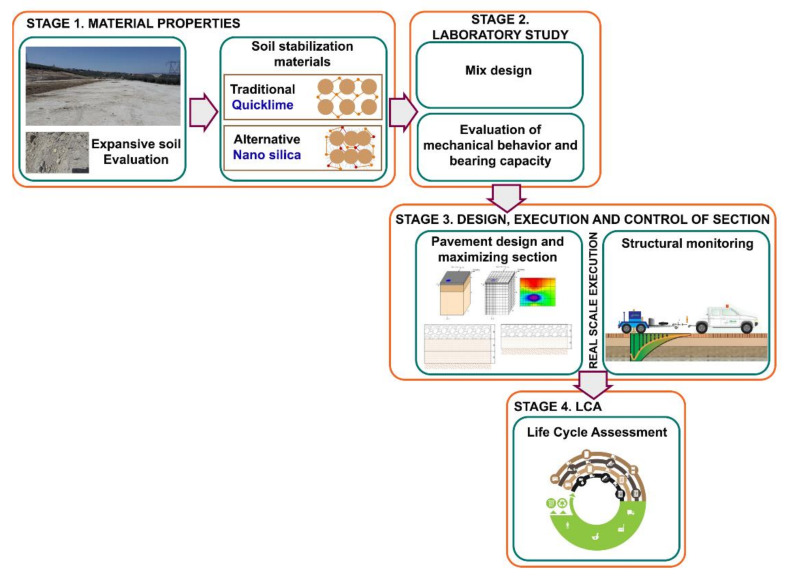
Experimental program.

**Figure 2 materials-13-03058-f002:**
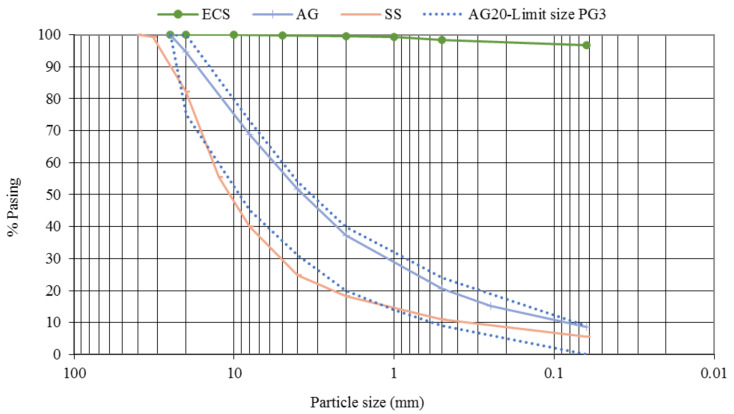
Particle size distribution curves.

**Figure 3 materials-13-03058-f003:**
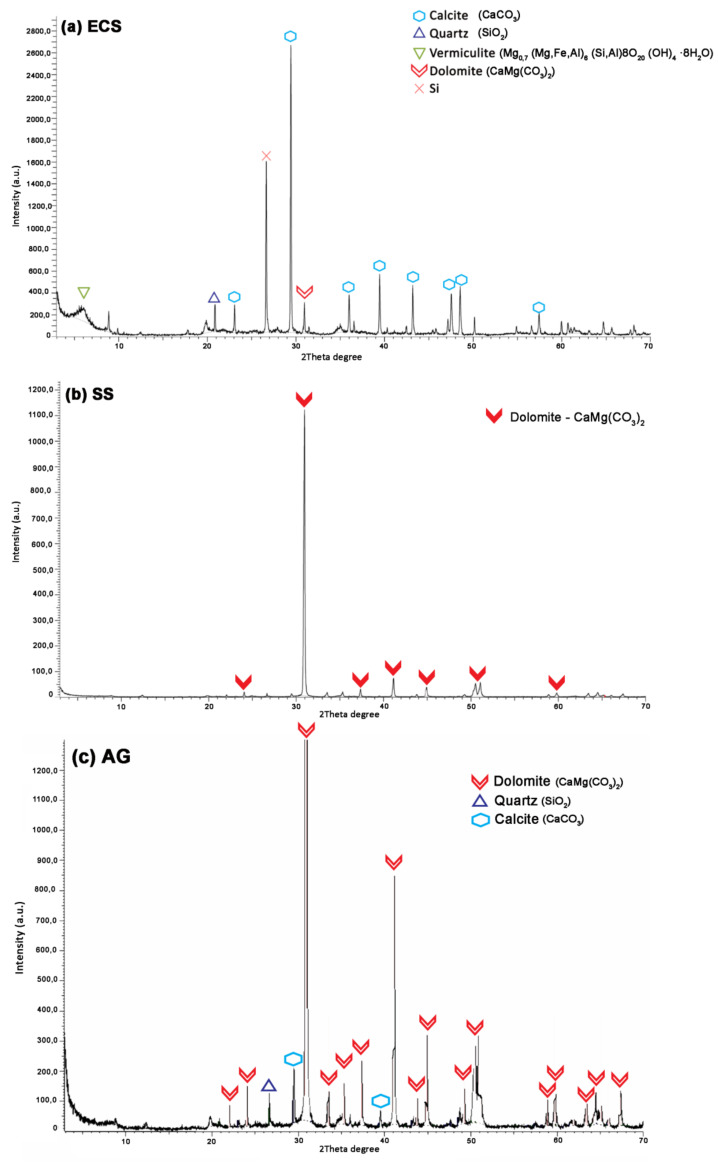
XRD patterns for samples of (**a**) ECS, (**b**) SS, (**c**) AG.

**Figure 4 materials-13-03058-f004:**
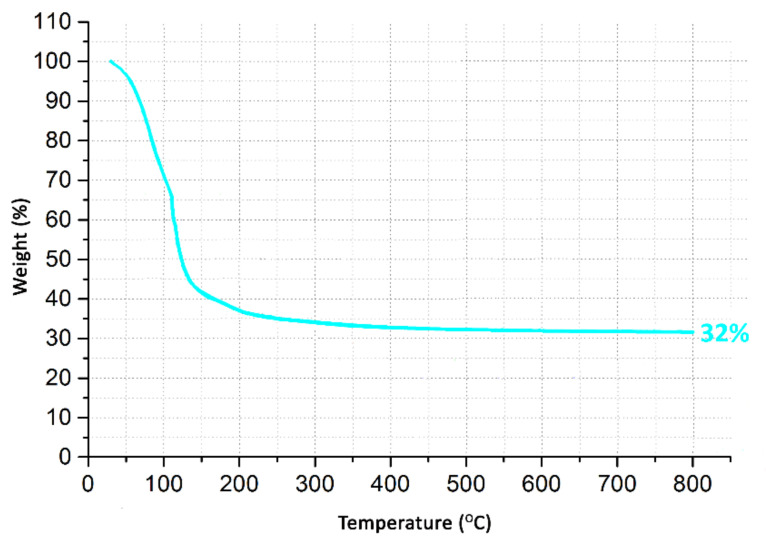
TGA curve of SSBA.

**Figure 5 materials-13-03058-f005:**
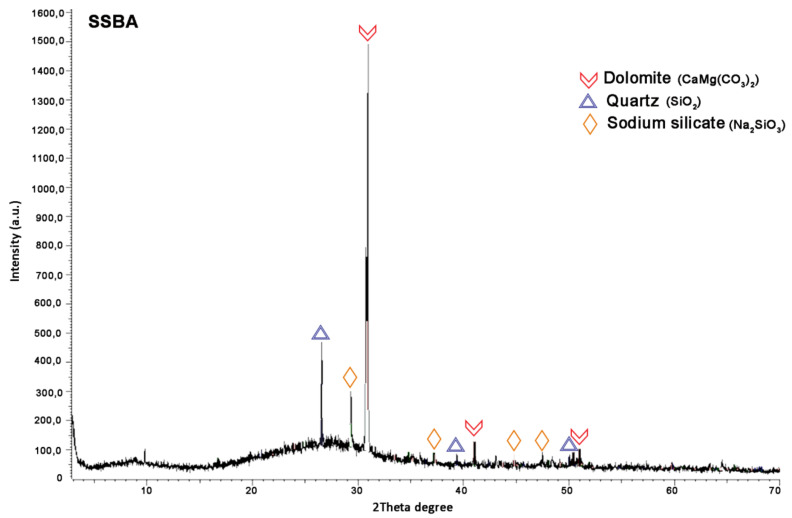
SSBA mineralogical composition by X-Ray diffraction (DRX).

**Figure 6 materials-13-03058-f006:**
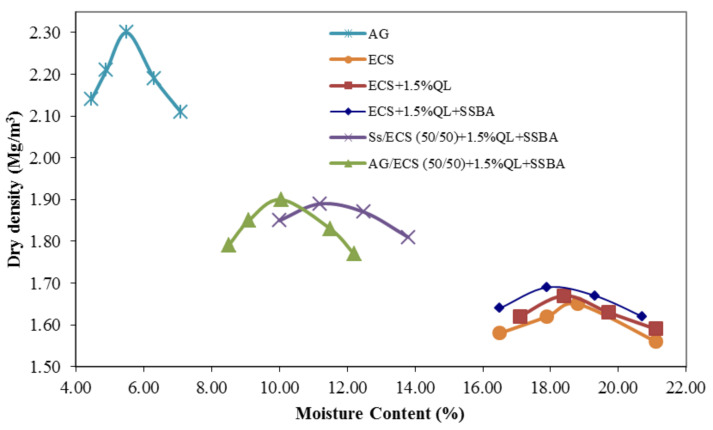
Moisture–density relationship.

**Figure 7 materials-13-03058-f007:**
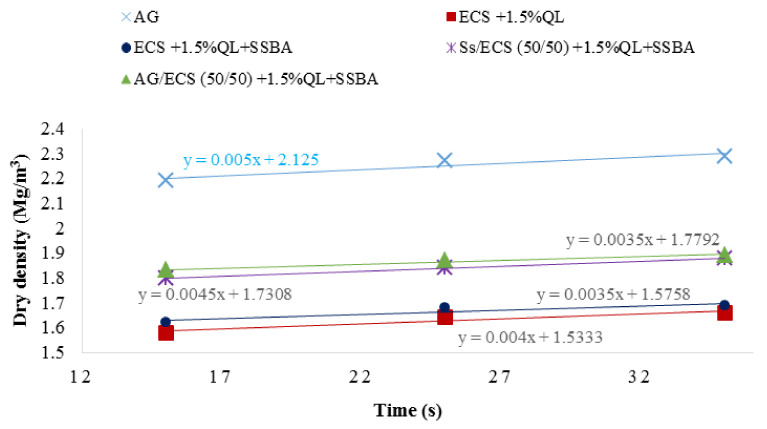
Dry density–compaction time relationship.

**Figure 8 materials-13-03058-f008:**
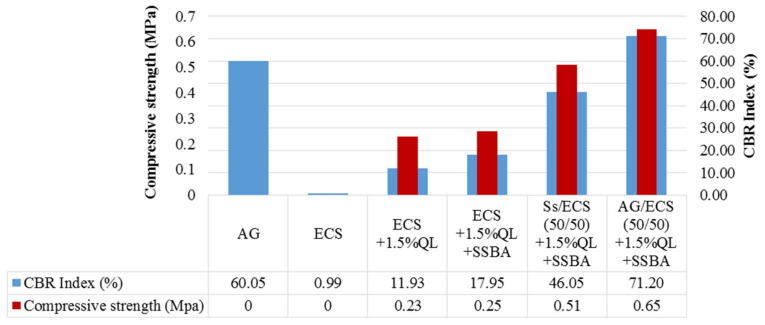
Results from the CBR index test and compressive strength test.

**Figure 9 materials-13-03058-f009:**
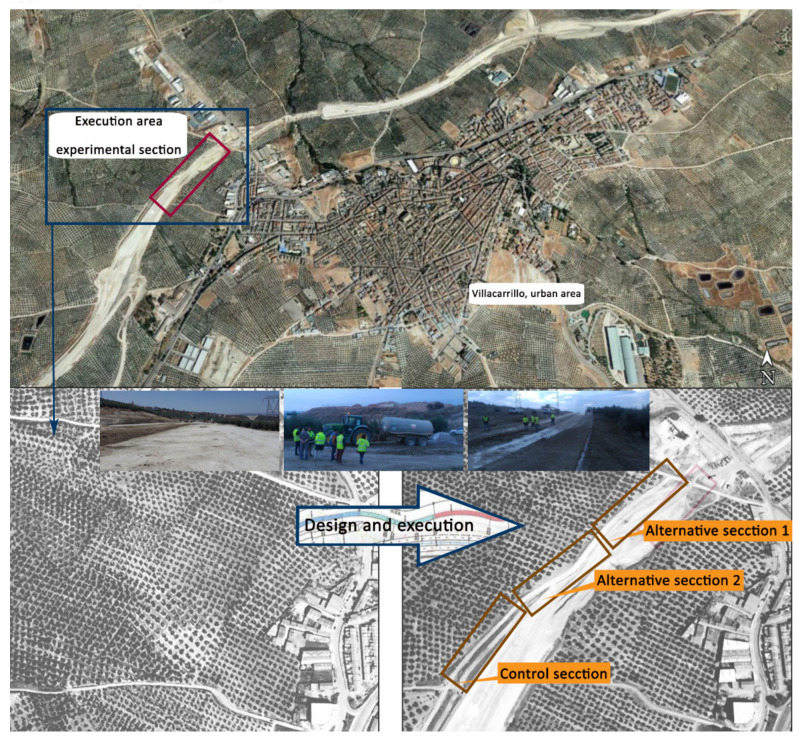
Execution area of the experimental section and location of each constructed alternative.

**Figure 10 materials-13-03058-f010:**
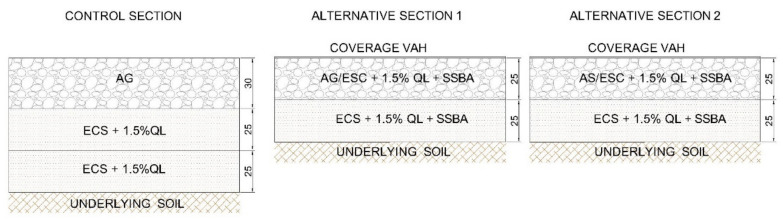
Sections implemented.

**Figure 11 materials-13-03058-f011:**
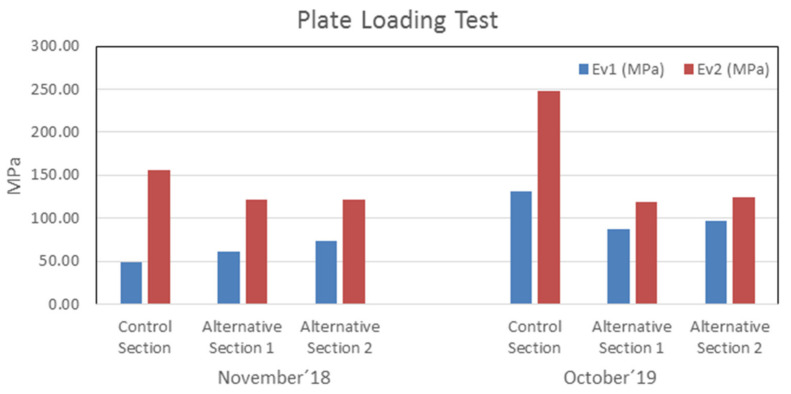
Value of plate loading test.

**Figure 12 materials-13-03058-f012:**
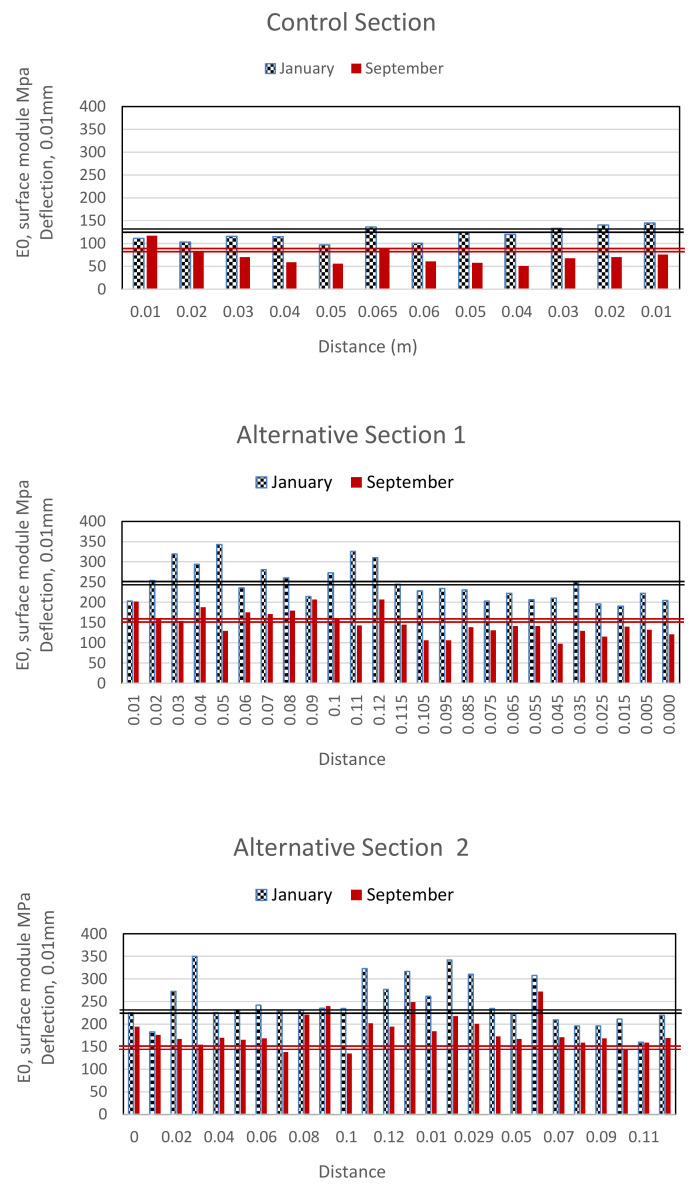
Deflection measurements.

**Figure 13 materials-13-03058-f013:**
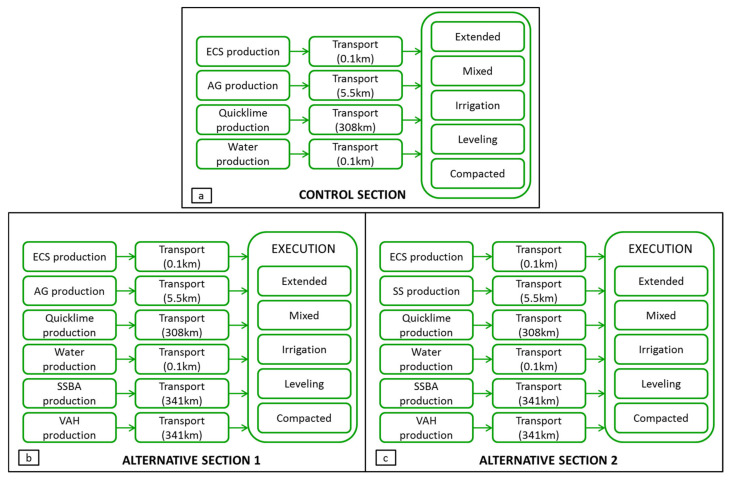
System boundaries: (**a**) Control Section; (**b**) Alternative section 1; (**c**) Alternative section 2.

**Figure 14 materials-13-03058-f014:**
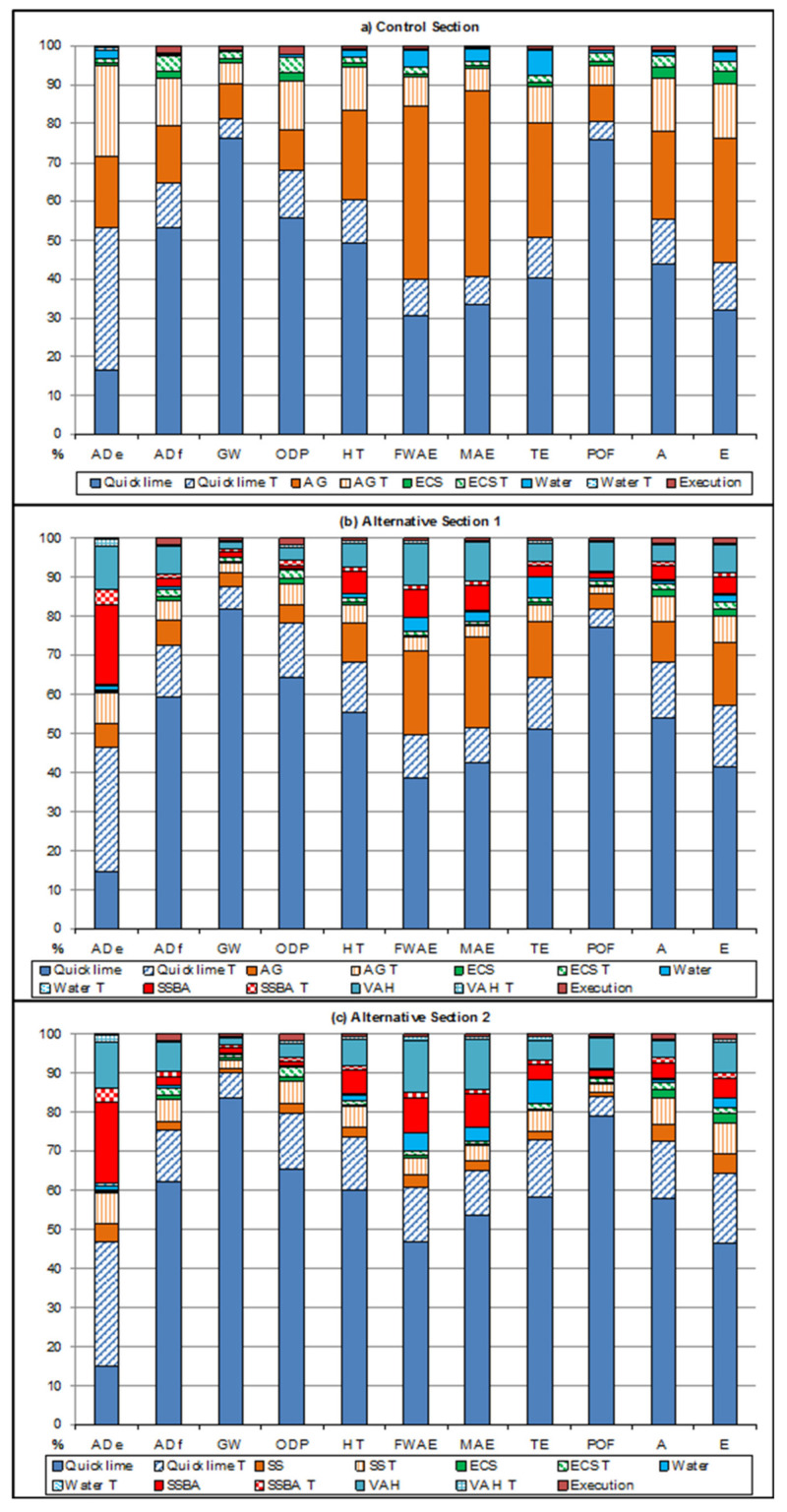
Contribution to the impact by processes: (**a**) Control Section; (**b**) Alternative section 1; (**c**) Alternative section 2.

**Table 1 materials-13-03058-t001:** Physical and chemical properties of Expansive clay soil (ECS), Artificial gravel (AG) and Selected soil (SS).

Properties	ECS	AG	SS	Required Limits—PG3		Test Method
				AG	SS	
**Atterberg limits**						UNE-EN 103103 UNE—EN 103104
Liquid limit	58	-	15.6	-	30	
Plastic limit	35.21	-	12.7	-	-	
Index plasticity	22.79	-	2.9	-	10	
**Density-SSD (kg/dm^3^)**						UNE-EN 1097-6:2014
0–4 mm	1.60	2.79	2.7			
4–31.5 mm	-	2.79	2.81			
**Water absorption (%)**						
0–4 mm	-	1.6	1.75			
4–31.5 mm	-	2.5	2.56			
**Sand equivalent (%)**	6	57	19	30	-	UNE-EN 933-8:2012
**Organic matter %**	0.27	<0.1	<0.1	-	0.2	UNE 103204:93
**Water-soluble sulphate (%SO_3_)**	<0.1	<0.1	<0.1	0.1	0.2	UNE-EN 1744-1
**Main Components XRF (%)**				-	-	
Na	0.15	0.05	0.15			
Mg	0.96	7.87	2.73			
Al	4.09	3.1	3.45			
Si	12.9	5.21	9.12			
P	0.04	0.02	0.03			
S	0.03	0.04	0.05			
Cl	-	-	0.02			
K	1.1	0.80	0.88			
Ca	18.8	20.9	21.3			
Ti	0.25	0.14	0.23			
Cr	-	-	0.05			
Mn	0.03	0.03	0.05			
Fe	1.84	1.06	1.68			
Sr	0.06	0.01	0.03			

**Table 2 materials-13-03058-t002:** Physical and chemical properties of SSBA.

Properties		SSBA
Density-Saturated Surface Dry (SSD) (kg/dm^3^)		1.4
Main Components XRF (%)		
Na		12.1
Mg		0.15
Al		0.182
Si		24.2
S		0.02
Cl		0.025
K		0.026
Ca		0.279
Fe		0.057

**Table 3 materials-13-03058-t003:** Dosages of the mixtures.

Nomenclature	Materials (kg/m^3^)					VHA (kg/m^2^) (Surface Irrigation)
	ECS	AG	SS	Quicklime (QL)	SSBA	
**ECS + 1.5%QL**	1655	-	-	25.00	-	-
**ECS + 1.5%QL +SSBA**	1664.65	-	-	25.35	0.910	-
**AG/ECS (50/50) + 1.5%QL + SSBA**	810.75	1135.75	-	28.50	0.910	0.200
**SS/ECS (50/50) + 1.5%QL + SSBA**	800.75	-	1085.75	28.35	0.910	0.200

**Table 4 materials-13-03058-t004:** Dry density and optimum moisture results.

	Dry Density (mg/m^3^)	Moisture (%)
AG	2.3	5.5
ECS	1.65	18.8
ECS+1.5%QL	1.67	18.4
ECS+1.5%QL+SSBA	1.69	17.9
SS/ECS (50/50) + 1.5%QL+SSBA	1.89	11.2
AG/ECS (50/50) + 1.5%QL+SSBA	1.90	10.05

**Table 5 materials-13-03058-t005:** In situ assessments of density and moisture.

Properties	ECS + 1.5%QL	AG	ECS + 1.5%QL + SSBA	AG/ECS + 1.5%QL + SSBA	SS/ECS + 1.5%QL + SSBA
**Density (kg/dm^3^)**	1.63	2.26	1.64	1.85	1.86
Mean					
**Compaction (%)**	97.3	98.3	97.3	97.4	98.7
Mean					
**Moisture content (%)**	20.5	6.64	19.23	10.89	12.21
Mean					

**Table 6 materials-13-03058-t006:** Ev_2_/Ev_1_ relationship for each section.

	ControlSection	Alternative Section 1	Alternative Section 2
November 2018			
R = Ev_2_/Ev_1_ (MPa)	3.21	1.96	1.64
October 2019			
R = Ev_2_/Ev_2_ (MPa)	2.63	1.36	1.29

**Table 7 materials-13-03058-t007:** Inverse module in the different layers.

Section	E_i1_ (MPa)	E_i2_ (MPa)	E_i3_ (MPa)
Control Section	316	145	71
Alternative 1	172	36	79
Alternative 2	188	44	66

E_i1_ (MPa): tread layer, E_i2_ (MPa):stabilized sub-base, E_i3_ (MPa):underlying natural terrain.

**Table 8 materials-13-03058-t008:** Inventory data for executed sections.

Consumption per FU	Control Section			Alternative Section 1			Alternative Section 2		
	t	h (OT^a^)	km	t	h (OT^a^)	km	t	h (OT^a^)	km
Raw material									
AG	729.948		5.5	303.6		5.5	-		-
SS	-		-	-		-	303.6		5.5
ECS	489.72		0.1	244.86		0.1	244.86		0.1
Quicklime	12.24		308	13.32		308	13.14		308
Water	186.48		0.1	123.20		0.1	127.38		0.1
SSBA	-		-	0.4368		341	0.4368		341
VAH	-		-	0.192		341	0.192		341
Machinery									
Rotavator		6			4.8			4.8	
Paning tractor		1.2			1.2			1.2	
Tank truck		0.432			0.3816			0.3816	
Motor grader		1.8			1.2			1.2	
Compactor		1.8			1.2			1.2	

OT^a^: Operating time.

**Table 9 materials-13-03058-t009:** Characterization results of sections.

Impact Category	Units	Control Section	Alternative Section 1	(∆%)	Alternative Section 2	(∆%)
ADe	kg Sb eq.	8.74 × 10^−3^	1.09 x 10^−2^	24.35	1.06 × 10^−2^	21.22
ADf	MJ	1.05 × 10^5^	1.02 × 10^5^	−3.18	9.65 × 10^4^	−8.22
GW	kg CO_2_ eq.	1.50 × 10^4^	1.51 × 10^4^	0.95	1.46 × 10^4^	−2.86
ODP	kg CFC-11 eq.	1.21 × 10^−3^	1.14 × 10^−3^	−5.63	1.10 × 10^−3^	−8.69
HT	kg 1.4-DB eq.	2.21 × 10^3^	2.13 × 10^3^	−3.75	1.95 × 10^3^	−11.96
FWAE	kg 1.4-DB eq.	9.82 × 10^2^	8.54 × 10^2^	−13.06	6.89 × 10^2^	−29.87
MAE	kg 1.4-DB eq.	3.48 × 10^6^	2.97 × 10^6^	−14.55	2.33 × 10^6^	−33.09
TE	kg 1.4-DB eq.	1.12 × 10^1^	9.58 × 10^0^	−14.11	8.34 × 10^0^	−25.27
POF	kg C_2_H_4_ eq.	3.09 × 10^0^	3.30 × 10^0^	6.88	3.18 × 10^0^	3.03
A	kg SO_2_ eq.	3.47 × 10^1^	3.07 × 10^1^	−11.56	2.84 × 10^1^	−18.32
E	kg PO_4_ eq.	7.93 × 10^0^	6.66 × 10^0^	−15.95	5.87 × 10^0^	−25.94

**Table 10 materials-13-03058-t010:** Impact assessment values by layer component for GW category (kg CO_2_ eq.).

Control Section	kg CO_2_ eq.	Alternative Section 1	kg CO_2_ eq.	Alternative Section 2	kg CO_2_ eq.
ECS + 1.5%QL	6180.2	ECS + 1.5%QL	6179.4	ECS + 1.5%QL	6179.4
ECS + 1.5%QL	6601.8	ECS/AG + 1.5%QL	8357.8	ECS/SS + 1.5%QL	7785.5
AG	2214.4	SSBA	279	SSBA	279
		VAH	323	VAH	323
**Total**	**14,996.4**	**Total**	**15,139.3**	**Total**	**14,566.9**

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
