# Peer review of "Use of Nanomaterials in the Stabilization of Expansive Soils into a Road Real-Scale Application"

_materials, 2020, doi:10.3390/ma13143058_

Round 1

Reviewer 1 Report

  1. This manuscript presents a study investigating soil stabilization using nano materials via laboratory testing and field application. In general, this topic is interesting and worth investigating, and the analysis methodology is sound in this study. The results and data presented are convincing. However, I strongly recommend a significant improvement on the technical writing of this paper.
  2. It seems to the reviewer that the title is not adequate enough, and some words in the title are redundant. Moreover, why is a period needed in the title?
  3. In order to help readers to better understand the background of this study, it is recommended to add more contents to elaborate the objectives and motivations in detail, especially it is needed to clarify what gap this study is bridging and the limitations in current state-of-the-art.
  4. Was soil suction measured in this study? If not, please explain why.
  5. How many replicates of samples were tested for each set of testing? Was there any statistical analysis being performed on the results? 
  6. How were the mixing contents of the stabilizers established? Please explain in detail, especially in terms of the control of the moisture contents.
  7. Is there anything else that the authors can interpret from the XRD test results?
  8. Table 4 is invisible in this PDF manuscript.
  9. Please consult with this journal office/editor in chief to see if you are allowed to use the commercial names of ISOVIA 7030/7350 in this scientific journal.
  10. It is recommended to significantly revise the conclusions. It is not necessary to repeat any results in this section, but could be more focused on the key findings, especially those could benefit future studies, so that the conclusions section can be more brief and concise. 

Author Response

Francisco Agrela Sainz

Construction Engineering
University of Córdoba
Campus de Rabanales

Ctra. Nacional IV, Km. 396
14014 CÓRDOBA- SPAIN

Dear Ms. Keria Zhao

Attached please find our revised manuscript materials-834853 entitled:

“USE OF NANOMATERIALS TO IMPROVE THE STABILIZATION OF EXPANSIVE SOILS.
FROM LAB TESTS TO REAL-SCALE APPLICATION”.

We are grateful for the constructive comments by yourself, which we believe have allowed us to substantially improve our manuscript.

We have thoroughly revised the manuscript. I would like to first address the principle concerns (in blue) and indicate how we have dealt with them. In addition, the manuscript with the changes made is attached (Track changes in word document). According to the comments made by the reviewers, the commercial names of the products used in the study have been removed

Reviewers' comments:

Reviewer #1:

It seems to the reviewer that the title is not adequate enough, and some words in the title are redundant. Moreover, why is a period needed in the title?

Thank you for your comment. The title has been changed  

In order to help readers to better understand the background of this study, it is recommended to add more contents to elaborate the objectives and motivations in detail, especially it is needed to clarify what gap this study is bridging and the limitations in current state-of-the-art.

Thank you for your comment. The introduction has been extended and the experimental program has been structured in a more easily understood way.

Was soil suction measured in this study? If not, please explain why.

Thank you very much for your comment. A detailed technical characterization of the materials was carried out in accordance with the Spanish General Technical Specifications for road construction (PG-3). PG-3 regulates the specifications that must be met by the materials that are part of the different profile sections of a road.

We will consider the suction test for future research.

How many replicates of samples were tested for each set of testing? Was there any statistical analysis being performed on the results? 

Three samples were made for each set of tests, the values ​​obtained for each set of tests were similar, therefore a statistical analysis was not introduced.

How were the mixing contents of the stabilizers established? Please explain in detail, especially in terms of the control of the moisture contents.

Thank you very much for your comment. The stabilizers were applied based on the dry weight of the material to be stabilized. In the laboratory, the soil moisture was taken and 1.5% lime was applied. For the dosage of the nanomaterial, knowing the density, 0.65 l/m3 was added to each mixture under study

Is there anything else that the authors can interpret from the XRD test results?

The XRD test was performed to see its main components of the materials under study, it was observed that the main component of the expansive clay soil was calcite, from the artificial gravel it was shown to be dolomite and the selected soil a mixture of dolomite and remains calcite.

Table 4 is invisible in this PDF manuscript.

We agree with your comment. In the document in "word" the table is correct. There was an error editing the PDF.

Please consult with this journal office/editor in chief to see if you are allowed to use the commercial names of ISOVIA 7030/7350 in this scientific journal.

Commercial names have been removed in the manuscript.

It is recommended to significantly revise the conclusions. It is not necessary to repeat any results in this section, but could be more focused on the key findings, especially those could benefit future studies, so that the conclusions section can be briefer and more concise. 

Thank you for your comment. The conclusions have been narrowed down. Showing only the most important aspects.

Reviewer 2 Report

The paper presents the physical properties and mechanical behavior of soils stabilized with silicate-based nanomaterial called ISOVIA 7030 and show the application in structural layers of rural roads. In this study three types of soils were analyzed: expansive clay soil, artificial gravel and selected soils, and two stabilizers were applied: lime and nano-silanes.

As overall the paper deals with an important topic and generally well written. However, I felt that the authors tried to include into the paper maximum of information, so from the certain point I started to feel completely lost. Firstly, I didn’t find a clear goal of the paper. Instead different goals were presented in the abstract, introduction and experimental program. Last paragraph of the introduction is rather conclusions than introduction into the paper.

Secondly, the authors tried to include in one paper laboratory and field experiments, and the life cycle assessment. On the one hand it is great but on the other hand it is easy to lose the main idea of the paper and also it is easy to feel lost already on the half way. Sometimes I had a feeling that different parts of the paper were written by different people.

Also I feel that paper is overloaded by different abbreviations. Would be nice to repeat time to time the full names again. Otherwise it is so difficult to follow.

Introduction is missing the background and state-of-the-art of similar laboratory and field investigations of using silicates in soil stabilization and what were main outcomes and what is special about research presented in the paper.

Experimental chapter has “Materials” part but not “Methods” part.

Life Cycle Assessment (LCA) chapter on the one hand has a lot of information but on the other hand more questions arise for the input parameters.

Discussion chapter is way too short, only half page. It should be more detailed. Some information from other chapters where the results are discussed can be moved here.

Conclusions are very well written and include all necessary summary.

More detailed comments:

Title.

In the title could you please put the colon between “Use of Nanomaterials to Improve the Stabilization of Expansive Soils.” And “ from Lab Tests to Real-Scale Application”

Abstract

Please give a full name for CBR index (The Californian Bearing Ratio (CBR))

Line 86-88 the aim is better to move in the end of the Introduction

Line 101 – where was the experimental site location? It should be mentioned here as well.

Line 107 it should be called “Materials and methods” and also should include a chapter “methods” where you will describe all methods you used in your studies

116 – was this real-scale study conducted for the research presented in the paper or for some other study? If for other study, do you have any references?

Line 123 – “General Technical Specification for Construction of Roads and Bridges PG-3” – do you have a reference for this document? Is it Spanish standard?

Line 126 in the title could you please put both the full name and the abbreviation

Table 1. What does it mean «NP» in the table?

It was not clear for me if “Required limits –PG3” – are the limits from the technical specification which you included into the table for comparison of your numbers with the numbers from specifications?

Line 131 “as selected soil soils” – should you leave only one “soil”?

Line 143-144 “a Cu twist tube” – is it part of some standard equipment for XRD?

Line 147-148 it would be beneficial if time to time you would give full names for the abbreviations (ex. ECS). Also if you give the names of the minerals, why did you leave just “Si” on this case? Names for the minerals should be with small letters but not the capital as it is now. Do you have a quotative analyze (what is the percentage of every mineral)?

Line 149-150 it was not understandable why suddenly “Guadalquivir River” was mentioned. Is it the location of test sites?

Line 164 “the chemical properties XRF” – please specify what did you mean. Did you perform a X-ray fluorescence? It should be mentioned before. Please specify what kind of equipment did you use

Line 169 “The progressive weight loss to 200 ºC is due 168 to the loss of water in the nano-silane” – could you please specify how did you make this conclusion?

 “ISOVIA 7030 showed high silica content, containing 32%.” – I thought that TGA shows the loss of water but not the amount of silica

Line 172-173 could you please describe better how did you come to this conclusions regarding nano-silica mass content?

Table 2. Could you please specify abbreviation and units for density-SSD (kg/dm3)

Line 189 please give the full name for the abbreviation « CBR index”

Table 4 please correct it because numbers are not very visible

Line 228 please give the full name for CBR and describe how do you calculate it and what kind of tests do you need for it in methods. You mentioned it 3 times before starting to describe it after the line 239.

Table 5. Please make sure that all text on the first line of the table is written in one line, otherwise It can be read differently

Line 371 did you mean the table 7?

Line 378 It would be better if you would make one single chapter on used methodology in the paper instead of describing the methods every time in separate chapters

Line 379 “assessment” should be with double “ss” in both places.

Line 400 what’s kind of “artificial gravel”? what is the difference with the natural one?

Line 429 what is “the CML-IA (v 4.7) method”

Line 467 please give again the full names for the abbreviations even if you mentioned them 2 pages ago.

Lines 454-455 I didn’t understand what is the meaning of this sentence: “Lime, hydrated, lose weight {RoW} market” and “Tap water {Europe without 454 Switzerland}, market”, respectively.”

Lines 463-526 I got completely lost in the text with so many abbreviations. Very difficult to follow and difficult to understand

Line 530 please expand the chapter Discussions.

Line 531 “the impacts generated”: the impacts of what and toward what? In the beginning it was described as “the environmental impact” but after that only “impacts” left.

Author Response

Francisco Agrela Sainz

Construction Engineering
University of Córdoba
Campus de Rabanales

Ctra. Nacional IV, Km. 396
14014 CÓRDOBA- SPAIN

Dear Ms. Keria Zhao

Attached please find our revised manuscript materials-834853 entitled:

“USE OF NANOMATERIALS TO IMPROVE THE STABILIZATION OF EXPANSIVE SOILS.
FROM LAB TESTS TO REAL-SCALE APPLICATION”.

We are grateful for the constructive comments by yourself, which we believe have allowed us to substantially improve our manuscript.

We have thoroughly revised the manuscript. I would like to first address the principle concerns (in blue) and indicate how we have dealt with them. In addition, the manuscript with the changes made is attached (Track changes in word document). According to the comments made by the reviewers, the commercial names of the products used in the study have been removed

Reviewers' comments:

Reviewer #2:

The paper presents the physical properties and mechanical behavior of soils stabilized with silicate-based nanomaterial called ISOVIA 7030 and show the application in structural layers of rural roads. In this study three types of soils were analyzed: expansive clay soil, artificial gravel and selected soils, and two stabilizers were applied: lime and nano-silanes.

As overall the paper deals with an important topic and generally well written. However, I felt that the authors tried to include into the paper maximum of information, so from the certain point I started to feel completely lost. Firstly, I didn’t find a clear goal of the paper. Instead different goals were presented in the abstract, introduction and experimental program. Last paragraph of the introduction is rather conclusions than introduction into the paper.

The work comprises four differentiated parts. A basic characterization of the material used to know its properties (Expansive clay soil, selected soil and nanomaterial). A second part in which different mixes of soil and nanomaterial are evaluated in the laboratory, by means of these tests the bearing capacity of the soil is checked. The tests carried out are those required by the Spanish PG-3 regulations for the evaluation of soils for the subsequent construction of traffic routes. A third test, in which a real road section is built using nanomaterial, facilitates the reduction of the thickness of the section. The properties acquired by the section are studied by means of deflection tests and loading and unloading plate, which are the two most common methods for evaluating a section by means of non-destructive tests. Finally, to evaluate the environmental impact caused by the execution of the real section alternative to a traditional construction solution with nanomaterials, a life cycle analysis is developed, which shows the reduction in consumption and CO2 emissions.

A section has been included in the experimental program that explains more clearly the phases developed in the study.

Secondly, the authors tried to include in one paper laboratory and field experiments, and the life cycle assessment. On the one hand it is great but on the other hand it is easy to lose the main idea of the paper and also it is easy to feel lost already on the half way. Sometimes I had a feeling that different parts of the paper were written by different people.

A study has been carried out to evaluate the real application of the use of nanomaterials in soil stabilization. For this reason, the authors considered it necessary to carry out a complete study, from laboratory tests to full-scale application. An attempt has been made to explain in more detail the study carried out and to unify it.

Also I feel that paper is overloaded by different abbreviations. Would be nice to repeat time to time the full names again. Otherwise it is so difficult to follow.

Thanks for the advice. Full names have been repeated throughout the text.

Introduction is missing the background and state-of-the-art of similar laboratory and field investigations of using silicates in soil stabilization and what were main outcomes and what is special about research presented in the paper.

Thank you for your comment. The introduction has been extended and the background to the use of silicates in soil stabilization has been reviewed more thoroughly.

Experimental chapter has “Materials” part but not “Methods” part.

Section 4 describes the methods and results obtained from the laboratory tests.

Life Cycle Assessment (LCA) chapter on the one hand has a lot of information but on the other hand more questions arise for the input parameters.

It is important to assess the environmental impact caused by the actual application of nanomaterials in the construction of a road section. There are no previous studies on LCA in an application of this type. For that reason, an extensive atonement of the developed method is necessary.

Discussion chapter is way too short, only half page. It should be more detailed. Some information from other chapters where the results are discussed can be moved here.

Thanks for the comment. The discussion section referred only to the LCA section, this has been clarified in the text. The discussion of the results obtained in the laboratory tests is shown in section 4 and that of the short and medium-term execution and monitoring tests in section 5. All these aspects have been clarified in the text. Conclusions are very well written and include all necessary summary.  

More detailed comments:

Title.

In the title could you please put the colon between “Use of Nanomaterials to Improve the Stabilization of Expansive Soils.” And “ from Lab Tests to Real-Scale Application

Thank you. The title has been changed.

Abstract

Please give a full name for CBR index (The Californian Bearing Ratio (CBR))

Thank you for your comment. It's been changed.

Line 86-88 the aim is better to move in the end of the Introduction

Thank you. It has been modified in the text.

Line 101 – where was the experimental site location? It should be mentioned here as well.

Thank you, the place of execution has been specified in the text.

Line 107 it should be called “Materials and methods” and also should include a chapter “methods” where you will describe all methods you used in your studies

Thank you. The article has been divided into sections as explained in the experimental programme: Stage 1 materials, in which their properties are explained, stage 2 laboratory study, in which the test methods and results are explained, stage 3 execution of the road section and full-scale tests, in which the methods and results obtained are explained, and stage 4 LCA, in which the methodology and results are again explained. Due to the length of the paper, the authors consider this order appropriate so as not to get lost when presenting the work.

116 – was this real-scale study conducted for the research presented in the paper or for some other study? If for other study, do you have any references?

An experimental road section was executed. It has been clarified in the text.

Line 123 – “General Technical Specification for Construction of Roads and Bridges PG-3” – do you have a reference for this document? Is it Spanish standard?

Thanks for your comment, the reference has been included. Line 126 in the title could you please put both the full name and the abbreviation

Thank you for your comment. The full name has been included in the title of the table

Table 1. What does it mean «NP» in the table?

The term NP has been changed to -. Considering that the material has no liquid or plastic limit.

It was not clear for me if “Required limits –PG3” – are the limits from the technical specification which you included into the table for comparison of your numbers with the numbers from specifications?

Yes, that's right. The limits introduced in the column "Required Limits" of the table are those established by the PG-3 regulation. The results we have obtained in the laboratory when analyzing the soil have been compared with these values.

Line 131 “as selected soil soils” – should you leave only one “soil”?

Thank you, it's been eliminated.

Line 143-144 “a Cu twist tube” – is it part of some standard equipment for XRD?

Yes, it's part of the XRD equipment.

Line 147-148 it would be beneficial if time to time you would give full names for the abbreviations (ex. ECS). Also if you give the names of the minerals, why did you leave just “Si” on this case? Names for the minerals should be with small letters but not the capital as it is now. Do you have a quotative analyze (what is the percentage of every mineral)?

Thank you. The name of the minerals has been included. The percentage of each in the XRD has not been specified as it has been shown by the XRF analysis.

Line 149-150 it was not understandable why suddenly “Guadalquivir River” was mentioned. Is it the location of test sites?

The area near the Guadalquivir River, located in the south of Spain, is the area where the project is located, exactly in Villacarrillo, Jaén. The soil     characteristics in that area are very similar. That is why it is referred to in the text.

Line 164 “the chemical properties XRF” – please specify what did you mean. Did you perform a X-ray fluorescence? It should be mentioned before. Please specify what kind of equipment did you use

Thank you for your comment. The equipment used has been specified in the text.

Line 169 “The progressive weight loss to 200 ºC is due 168 to the loss of water in the nano-silane” – could you please specify how did you make this conclusion?

 “ISOVIA 7030 showed high silica content, containing 32%.” – I thought that TGA shows the loss of water but not the amount of silica

Line 172-173 could you please describe better how did you come to this conclusions regarding nano-silica mass content?

The three questions above are answered on the basis of the same argument. The ISOVIA 7030 product consists of a silicate solution, as can be seen in the following link: (https://www.acae.es/catalogos/cemex/fiebdc/ft-Isovia-7030-EN.pdf)

This is demonstrated based on the results of XRF and XRD (basically it could be described as a solution of sodium silicate nanoparticles). The analysis of TGA reveals two aspects:

(1) That it is dissolved in water, as the weight loss occurs below 150 ºC, the characteristic temperature where all the evaporation of the aqueous solvent takes place, as well as the water molecules physosorbide in the silicate itself, which could correspond to the last stage of weight loss (between 130-200ºC) If the solvent is not water but a liquid polymer of an organic nature, the weight loss should occur in the range of 200-300°C.

(2) According to the TGA, once it achieves a temperature of 200ºC, the weight of the material remains stable until the end of the test (at 800ºC). Only a minimal weight loss (1-2%) is barely detectable in this range. This fact confirms that the material that the solute making up the solution is a silicon-based compound that does not oxidize or decompose in this temperature range. The thermal behavior of silicates in the presence of water has been reported and is in agreement with our observations [https://doi.org/10.1016/j.bsecv.2019.06.004]. Therefore, the presence of silicates as the basis of the solution would be fully confirmed.

Table 2. Could you please specify abbreviation and units for density-SSD (kg/dm3)

It has been included in the table.

Line 189 please give the full name for the abbreviation « CBR index”

It has been included.

Table 4 please correct it because numbers are not very visible

The conversion to PDF of the document has caused table 4 not to be visible clearly. The original document is correct; we consider that it has been an editing problem.

Line 228 please give the full name for CBR and describe how do you calculate it and what kind of tests do you need for it in methods. You mentioned it 3 times before starting to describe it after the line 239.

The procedure for carrying out the CBR is defined in the UNE 103-502 standard. In addition, section 4.4 explains the method.

Table 5. Please make sure that all text on the first line of the table is written in one line, otherwise It can be read differently
Thank you, it's been modified.

Line 371 did you mean the table 7?

Thank you, it's been modified.

Line 378 It would be better if you would make one single chapter on used methodology in the paper instead of describing the methods every time in separate chapters

The sections of the paper have been restructured. It is divided into the 4 parts exposed at the beginning (Experimental Program) and exposing the methodology and results of each one of the parts separately to make the reading and understanding of the document easier.

Line 379 “assessment” should be with double “ss” in both places.

Thank you, it's been corrected.

Line 400 what’s kind of “artificial gravel”? what is the difference with the natural one?

The difference between both materials is defined in the Spanish PG-3 regulations. According to what is established in this standard, the use of artificial gravel is more appropriate for the execution of road sections, that is the main reason for its use.

According to the regulations, the difference between both types of soil is as follows:

Artificial gravel is a mixture of aggregates, totally or partially crushed, in which the particle size distribution of all the elements that make it up is of a continuous type.

Natural gravel is defined as material formed by uncrushed aggregates, granular soils, or a mixture of both.

Line 429 what is “the CML-IA (v 4.7) method”

The impact assessment performed by means of the LCA can be carried out according to different databases.

CML-IA is a database that contains characterisation factors for life cycle impact assessment (LCIA).

Line 467 please give again the full names for the abbreviations even if you mentioned them 2 pages ago.

Full names have been included for better understanding of the text.

Lines 454-455 I didn’t understand what is the meaning of this sentence: “Lime, hydrated, lose weight {RoW} market” and “Tap water {Europe without 454 Switzerland}, market”, respectively.”

It refers to the sections within the database used to enter the inventory of elements used in the execution of the road section, particularly lime and water.

It has been removed from the text, as the authors have considered it to be an excess of information that does not contribute to the document.

Lines 463-526 I got completely lost in the text with so many abbreviations. Very difficult to follow and difficult to understand

Thank you, full names have been included to make it easier to read.

Line 530 please expand the chapter Discussions.

As mentioned above, the discussion section refers only to the LCA. Each of the 4 parts of which this study is composed are discussed in the corresponding section. The name of the section has been changed so as not to confuse the reader.

Line 531 “the impacts generated”: the impacts of what and toward what? In the beginning it was described as “the environmental impact” but after that only “impacts” left.

Thank you, it's been corrected. Reference is made to environmental impact

Reviewer 3 Report

The manuscript is well structured and some experimental methods are well-conducted and reviewed. I think the paper can be of interest to the readers of Materials. However, the results of this manuscript should be verified and confirmed. I suggest to the authors should have a minor revision and the suggestions are as follows:

- Please merge the paragraph of the introduction and highlight the research.

- How many specimens are prepared in this study? Author should explain it.

- Table 4 is not clear. Please modify the quality of Table 4.

- Page 4; Line 106-110: please cite the references for the distribution of variables in rheology for the model concrete.

- For the discussion of lab. test and in-site test, each section also needs a better comparison of the stabilization. The author can cite the references for the comparison of nanomaterials and traditional materials.

- Please simplify the main conclusions from the research.

- Please indicate the roles and importance of the nanomaterials to improve the stabilization of expansive soils. Reader will interest this topic.

Author Response

Francisco Agrela Sainz

Construction Engineering
University of Córdoba
Campus de Rabanales

Ctra. Nacional IV, Km. 396
14014 CÓRDOBA- SPAIN

Dear Ms. Keria Zhao

Attached please find our revised manuscript materials-834853 entitled:

“USE OF NANOMATERIALS TO IMPROVE THE STABILIZATION OF EXPANSIVE SOILS.
FROM LAB TESTS TO REAL-SCALE APPLICATION”.

We are grateful for the constructive comments by yourself, which we believe have allowed us to substantially improve our manuscript.

We have thoroughly revised the manuscript. I would like to first address the principle concerns (in blue) and indicate how we have dealt with them. In addition, the manuscript with the changes made is attached (Track changes in word document). According to the comments made by the reviewers, the commercial names of the products used in the study have been removed

Reviewers' comments:

Reviewer #3:

Please merge the paragraph of the introduction and highlight the research.

Thank you, it's been modified

How many specimens are prepared in this study? Author should explain it.

Thank you very much for your comment, the authors have included the number of samples that were made for each set of tests.

“In this section, the results of the laboratory tests that were carried out on the mixtures that formed the different layers of the road sections are shown. The Modified Proctor compaction test, CBR index and unconfined compressive strength were determined. Three samples were made for each set of tests.”

Table 4 is not clear. Please modify the quality of Table 4.

We agree with your comment. In the document in "word" the figure looks correct. There was an error editing the PDF.

Page 4; Line 106-110: please cite the references for the distribution of variables in rheology for the model concrete.

I think your indication is not in accordance with the manuscript.

For the discussion of lab. test and in-site test, each section also needs a be tter comparison of the stabilization. The author can cite the references for the comparison of nanomaterials and traditional materials.

Thanks to your contribution, the discussion of the results has been extended and new references have been added

Please simplify the main conclusions from the research.

Thank you for your comment. The conclusions have been narrowed down. Showing only the most important aspects.

Please indicate the roles and importance of the nanomaterials to improve the stabilization of expansive soils. Reader will interest this topic.

Thank you very much for your comment, we have highlighted the importance of the nanomaterial:

“According to different studies [28,29] the use of nano-SiO2 shows an increase in unconfined compressive strength of the soil in relation to the use of other traditional stabilisers such as lime or cement.”

  1. Bahmani, S.H.; Huat, B.B.; Asadi, A.; Farzadnia, N. Stabilization of residual soil using SiO2 nanoparticles and cement. Construction and Building Materials 2014, 64, 350-359, doi:https://doi.org/10.1016/j.conbuildmat.2014.04.086.
  2. Haeri, S.M.; Hosseini, A.M.; Shahrabi, M.M.; Soleymani, S. Comparison of strength characteristics of Gorgan loessial soil improved by nanosilica, lime and Portland cement. In Proceedings of 15th Pan American Conference on Soil Mechanics and Geotechnical Engineering.DOI: 10.3233/978-1-61499-603-3-1820

Reviewer 4 Report

Manuscript presents very interesting study regarding use of nano-materials in road construction, but some improvements must be implemented, especially in the presentation of the results and analysis of results. Detailed remarks and questions are given below:

  1. Line 68 – there are a lot of materials where nano-materials have not been used, not only soil stabilization
  2. Line 71 and further – please change it into Aim and scope subsection. Also do not give conclusions in the introduction section
  3. Experimental program section need to be improved (especially graph). It should presents all tested variants and tests applied to them.
  4. Please add information regarding the nano-particles (like size, etc)
  5. Please add information regarding the numer of specimen. Also please show results scatter on the results graphs.
  6. Please improve the readability of the tables (especially 4). Do not use different size of fonts in the manuscript
  7. Is the 7 days soak typical for CBR method in Spain?
  8. Fig 10. How the reduction of the thickness determined?
  9. Why in steel plate tests so big difference in control section results between two tests and alternative sections did not show much differences? Especially in E2 result?
  10. Please add information on FWD backanalysis. Please describe in details how the thicknesses were determined. Why E2 results are so low? Which quartile was used in the analysis?

Author Response

Francisco Agrela Sainz

Construction Engineering
University of Córdoba
Campus de Rabanales

Ctra. Nacional IV, Km. 396
14014 CÓRDOBA- SPAIN

Dear Ms. Keria Zhao

Attached please find our revised manuscript materials-834853 entitled:

“USE OF NANOMATERIALS TO IMPROVE THE STABILIZATION OF EXPANSIVE SOILS.
FROM LAB TESTS TO REAL-SCALE APPLICATION”.

We are grateful for the constructive comments by yourself, which we believe have allowed us to substantially improve our manuscript.

We have thoroughly revised the manuscript. I would like to first address the principle concerns (in blue) and indicate how we have dealt with them. In addition, the manuscript with the changes made is attached (Track changes in word document). According to the comments made by the reviewers, the commercial names of the products used in the study have been removed

Reviewers' comments:

Reviewer #4:

Manuscript presents very interesting study regarding use of nano-materials in road construction, but some improvements must be implemented, especially in the presentation of the results and analysis of results. Detailed remarks and questions are given below:

Line 68 – there are a lot of materials where nano-materials have not been used, not only soil stabilization

Thank you for your comment. The authors consider that this idea is expressed in the text.

Line 71 and further – please change it into Aim and scope subsection. Also do not give conclusions in the introduction section

Thank you for your comment. The text has been reworked. By removing the conclusions and objectives from the introduction section.

Experimental program section need to be improved (especially graph). It should presents all tested variants and tests applied to them.

Thank you for your input. The experimental programme has been modified and explained more clearly. The authors have modified Figure 1, however, they consider that including all the tests in the figure would imply too much information. They consider that the clarification given in the text is sufficient.

Please add information regarding the nano-particles (like size, etc)

The nanomaterial used is a commercial product. The data shown in the paper is that which the company has allowed us to display. A link to the product's commercial data sheet is attached: (https://www.acae.es/catalogos/cemex/fiebdc/ft-Isovia-7030-EN.pdf)

Please add information regarding the numer of specimen. Also please show results scatter on the results graphs.

Thank you for your comment. The number of specimens made has been indicated in section 4.1

Please improve the readability of the tables (especially 4). Do not use different size of fonts in the manuscript

We agree with your comment. In the document in "word" the table is correct. There was an error editing the PDF. The font has been unified in the tables, sorry for the mistake.

Is the 7 days soak typical for CBR method in Spain?

Yes, according to the PG-3 standard for evaluation of stabilized soils. As well as in accordance with the UNE 13286-51

Fig 10. How the reduction of the thickness determined?

According to the results of compression strength and CBR obtained in the laboratory tests, a study of the determination of the elastic modulus was carried out. With these data, the calculations for the different pavement layers were carried out according to the fatigue limits for different thicknesses using the Everstress calculation program. The thicknesses that resulted in theoretical deflections and equivalent modules lower or similar to the control were considered for the dimensioning of the experimental section.

Why in steel plate tests so big difference in control section results between two tests and alternative sections did not show much differences? Especially in E2 result?

In the particular case of the control section, it had a notable increase in October (2019) compared to November (2018) due to the circulation of heavy vehicles on that section.

Due to the circulation of vehicles in this section, the layers that make up the road have been overcompacted, producing differential settlements over time and withstood heavy traffic.

In order to define the structure of the pavement in each case, three esplanade categories are established, named respectively E1, E2 and E3. These categories are determined according to the compressibility modulus in the second load cycle (Ev2), obtained in accordance with NLT-357 "Plate load test", the values of which are shown in the following table

ESPLANADE CATEGORY

E1

E2

E3

EV2

>60

>120

>300

All our results exceed 120 MPa required by Spanish regulations for the E2 traffic category.

Please add information on FWD backanalysis. Please describe in details how the thicknesses were determined. Why E2 results are so low? Which quartile was used in the analysis?

A Dynatest Heavy Weight Deflectometer 8081 equipped with seven geophones was used. The geophones were located at 0–300–450–600–900–1200–1500 mm.

The standard that regulates these loads and configurations is the “Technical Specifications for High-Performance Dynamic Monitoring Tests” [Public Works Agency of the Regional Government of Andalusia, PPTG ADAR. General technical specifications for high-performance dynamic monitoring tests, 2004. 

http://www.aopandalucia.es/inetfiles/area_tecnica/Calidad/ADAR/pliego_prescripciones_ADAR.pdf] from the Civil Works Agency of Regional Government of Andalusia (Spain).

The results obtained are within the ratio for an E2 traffic category according to Spanish regulations. Theoretical deflection calculated with multilayer software BISAR. This software applies different theories (Burmister, and Acum and Fox, and Shiffmann) implements a solution to determine tension and stress. The theoretical deviation was obtained for each layer and section according to the moduli of elasticity and Poisson ratios.

After complete the revision process, we hope that the revised manuscript does now fully meet the criteria and conditions for publication in Materials. Thank you very much for your efforts concerning our manuscript.

Yours sincerely,

Ph D. Francisco Agrela

Construction Engineering. University of Cordoba.

Round 2

Reviewer 2 Report

Dear Authors,

please find the comments in the attached file.

Author Response

Reviewers' comments:

Reviewer #2:

Line 86. Experimental program. In authors reply to the comments you describe very well your experimental program as a nice flow: “Text”.

I think you should include this part either in the abstract or include directly into the text from the line 86. You made a very nice summary which deserves to be a part of the paper. I found that this text (above) sounds much better in term of the language and “a flow” than what is now in the paper.

Thank you very much for your comments. The text has been included in the manuscript

Line 97. I don’t think that the word “auscultation” is the right word here since it means “the action of listening to sounds from the heart, lungs, or other organs, typically with a stethoscope, as a part of medical diagnosis.”

Thank you very much for your comment. It has been corrected in the text.

Line 105. Maybe it is better to add “Spain” after Andalusia since someone outside of Europe might read your paper and might not be aware about the location.

Thank you very much. You're right, it' s been included

Figure 1. The figured was improved a lot by adding the stages of the experimental programme. Now it looks much more clear. The only comment: instead of “lab study” it is better to give a full name “laboratory study”

Thank you, the figure has been modified

Line 173-176 I would recommend to add into the paper some of the explanations you gave me in your “reply to the comments”. I am not sure why did you choose not to do so since it was very well  explained and could help for other readers to understand it as well. I know that you gave the reference to another scientific paper. I looked at it but believe me that it was not so straight forward to connect with your data. I would recommend to incorporate at least some of the explanations into your text since it gives very good explanations to your results.

Thank you for your comment. The comments on the revision have been added to the manuscript.

Line 180 please add space between «SSBA” and “mass”

Thank you. It's been corrected.

Line 199-200 “Three specimens of each mixes were manufactured to carry out the tests-“ to which specimens from the Table 3 you referred in your text?

From each of the 4 mixtures, 3 specimens have been made to obtain an average value of the results obtained in the tests.

Table 4. Numbers were again not readable

Thank you, it's been corrected.

Line 249 “aggregate” – should be plural since they are many – aggregates

Thank you, it's been corrected.

Line 272-273 “On the contrary, expansive soils such as clays are considered non-usable soils. According to Spanish regulations.” – should it be one sentence?

Thank you, it's been corrected.

Line 281-282 “only 30% increases in the CBR index were achieved with respect to unstabilized soil.” But in the Conclusions, line 572, it stated “The addition of SSBA to expansive clay soil with lime increases the CBR index by 50%.” How could you explain this difference?

The results of the 30% CBR increase written in the manuscript refer to previous studies. The 50% increase was the result obtained in our work. It has been explained in the text. The difference between previous results and those obtained in the study may be due to the silicate base of the nanomaterial used.

Table 5. You could move “SS” to the next line, so you will have “SSBA” together

Thank you, it was a PDF editing problem

Line 329 “The use of SSBA resulted in a reduction of the optimum moisture values.” – do you have explanation why?

Thank you for your question. We are currently researching the influence of the addition of nanomaterials on the optimal moisture and density acquired by the soil. A microscopic study is being carried out to observe the internal reactions that occur in the mixes.  I am sorry I cannot answer your question, as it is the subject of the next study we are developing.

Reviewer 4 Report

Dear authors,

thank you very much for your detailed answers and implementing most of them in the manuscript. But two of the previous remarks can still be improved.

as for fig 10. where there are constructions for field section. You have given me the answer in separate file. Thank you for that, but I also think, that it would be also very interesting for the readers. So please put some information regarding design into the manuscript.

regarding FWD results. Your answer partially satisfies me. This kind of test, give very big scatter of results, due to the used algorythm (like in figure 12 for deflections). So for obtaining one number one need to chose a quartile or similar value. In your manuscript, is the fwd backcalculated value a mean value or some qurtile of values? You can describe it, or add a similar figure for modulus like is presented for deflections.

Also fig 12 need to have all information described - what are the lines on the figure.

Author Response

Reviewers' comments:

Reviewer #4:

Thank you very much for your detailed answers and implementing most of them in the manuscript. But two of the previous remarks can still be improved.

As for fig 10. where there are constructions for field section. You have given me the answer in separate file. Thank you for that, but I also think, that it would be also very interesting for the readers. So please put some information regarding design into the manuscript.

Thank you, the explanation has been added in the text.

Regarding FWD results. Your answer partially satisfies me. This kind of test, give very big scatter of results, due to the used algorythm (like in figure 12 for deflections). So for obtaining one number one need to chose a quartile or similar value. In your manuscript, is the fwd backcalculated value a mean value or some quartile of values? You can describe it, or add a similar figure for modulus like is presented for deflections

Thank you very much for your comment. The values shown in figure 12 are unit values (vertical lines) in each of the sections under study. Subsequently, a horizontal line has been drawn with the average deflection of the data set that makes up the section. I have included a clarification in the text "The mean deflection is shown with a black horizontal line for the test carried out in January, and a red horizontal line for the test carried out in September"

Also fig 12 need to have all information described - what are the lines on the figure

Thank you very much. We consider that the question has been answered in the previous question
